# Gut microbial co-abundance networks show specificity in inflammatory bowel disease and obesity

Lianmin Chen [1,2], Valerie Collij[1,3,14], Martin Jaeger[4,14], Inge C. L. van den Munckhof [4], Arnau Vich Vila [1,3], Alexander Kurilshikov [1], Ranko Gacesa [1,3], Trishla Sinha [1], Marije Oosting[4], Leo A. B. Joosten [4,5], Joost H. W. Rutten[4], Niels P. Riksen [4], Ramnik J. Xavier [6,7,8,9], Folkert Kuipers[2,10], Cisca Wijmenga[1,11], Alexandra Zhernakova[1], Mihai G. Netea [4,12,13], Rinse K. Weersma[3] & Jingyuan Fu [1,2 ✉]

The gut microbiome is an ecosystem that involves complex interactions. Currently, our knowledge about the role of the gut microbiome in health and disease relies mainly on differential microbial abundance, and little is known about the role of microbial interactions in the context of human disease. Here, we construct and compare microbial co-abundance networks using 2,379 metagenomes from four human cohorts: an inflammatory bowel disease (IBD) cohort, an obese cohort and two population-based cohorts. We find that the strengths of 38.6% of species co-abundances and 64.3% of pathway co-abundances vary significantly between cohorts, with 113 species and 1,050 pathway co-abundances showing IBD-specific effects and 281 pathway co-abundances showing obesity-specific effects. We can also replicate these IBD microbial co-abundances in longitudinal data from the IBD cohort of the integrative human microbiome (iHMP-IBD) project. Our study identifies several key species and pathways in IBD and obesity and provides evidence that altered microbial abundances in disease can influence their co-abundance relationship, which expands our current knowledge regarding microbial dysbiosis in disease.

[1] Department of Genetics, University of Groningen, University Medical Center Groningen, Groningen, the Netherlands. [2] Department of Pediatrics, University of Groningen, University Medical Center Groningen, Groningen, the Netherlands. [3] Department of Gastroenterology and Hepatology, University of Groningen, University Medical Center Groningen, Groningen, the Netherlands. [4] Department of Internal Medicine and Radboud Institute for Molecular Life Sciences, Radboud University Medical Center, Nijmegen, the Netherlands. [5] Department of Medical Genetics, Iuliu Hatieganu University of Medicine and Pharmacy, Cluj-Napoca, Romania. [6] Center for Computational and Integrative Biology, Massachusetts General Hospital, Boston, MA, USA. [7] Broad Institute of MIT and Harvard, Cambridge, MA, USA. [8] Gastrointestinal Unit and Center for the Study of Inflammatory Bowel Disease, Massachusetts General Hospital and Harvard Medical School, Boston, MA, USA. [9] Center for Microbiome Informatics and Therapeutics, Massachusetts Institute of Technology, Cambridge, MA, USA. [10] Department of Laboratory Medicine, University of Groningen, University Medical Center Groningen, Groningen, the Netherlands. [11] University of Groningen, Groningen, the Netherlands. [12] Department for Genomics & Immunoregulation, Life and Medical Sciences Institute, University of Bonn, 53115 Bonn, Germany. [13] Human Genomics Laboratory, Craiova University of Medicine and Pharmacy, Craiova, Romania. [14] These authors contributed equally: Valerie Collij, Martin Jaeger. ✉email: j.fu@umcg.nl

The human gut harbours a diverse community of micro-organisms that interact closely with both the host and each other. Gut microorganisms are involved in digestion and degradation of nutrients, maintenance of digestive tract integrity, stimulation of the host immune system and modulation of the host metabolism[1–5]. In recent years, associations have been identified between gut microbiome composition and the development of certain human diseases, including diabetes[6,7], cardiovascular disorders[8,9], obesity[10,11] and chronic gastrointestinal disorders like inflammatory bowel disease (IBD)[12–14]. Most associations to human diseases have been linked to lower microbial diversity, altered microbial composition and differing abundances of certain microorganisms and pathways[3,8,15–19]. However, the gut microbiome is an ecosystem in which microbes can exchange or compete for nutrients, signalling molecules or immune-evasion mechanisms through complicated ecological interactions that are far from fully understood[20–22]. Enthusiasm has thus been rising to decipher these microbial interactions in order to detect key microbes in health and disease[23,24]. One way of doing this is to create co-abundance networks based on correlations, a method that has the potential to study interactions between microbes and thereby generate hypotheses for experimental validation at a later stage[23,24].

Various network inference tools have been developed[25–29] and applied to infer microbial taxonomic networks in healthy individuals and in individuals with extreme longevity, gestational diabetes, Crohn's disease and colorectal cancer[30–35]. These studies have identified microbial genera that are potentially key in health and disease, e.g. *Porphyromonas* and *Bacteroides* in gestational diabetes[33]. However, these previous studies were either based on 16S rRNA sequencing data, which yields limited information on microbial species and pathways, or carried out in small cohorts[30–34]. A further limitation of 16S sequencing is that it can only identify microbial networks up to genus level. As different bacterial species can have very different functional properties, analysis at genus level cannot fully capture the biochemical interactions between microbes. In consequence, the importance of metabolic network construction from metagenomics data has recently been highlighted[24,36,37].

Here we present a metagenomics-based network analysis for bacterial species and metabolic pathways in 2379 individuals from 4 cohorts from the Netherlands (Supplementary Fig. 1): an IBD cohort ($n = 496$), an obesity cohort (300 Obesity cohort (300OB; $n = 298$), and 2 population-based cohorts (Lifelines-DEEP (LLD; $n = 1135$) and 500 Functional Genomics (500FG; $n = 450$)). We compare the microbial taxonomic and functional networks under different host health conditions and identify potential key species and pathways that shape host-associated microbial networks (Fig. 1). We find that the microbial species and pathway co-abundances vary significantly between cohorts and report IBD- and obesity-specific co-abundance networks, which expand our current knowledge regarding microbial dysbiosis in disease. The network key species and pathways identified in IBD and obesity highlight their potential roles in regulating the microbial ecosystem in disease.

## Results

**Construction of gut microbial co-abundance networks**. Metagenomic data of the 2379 participants from the four cohorts was processed using the same pipeline. Principle coordinate analysis showed that microbial composition and functional profiles are largely overlapped, although we observed a significant shift in species composition in the IBD cohort (Supplementary Fig. 2). A total of 134 bacterial species and 343 microbial pathways that were present in >20% of the samples in at least one cohort were included for microbial network inference. We established microbial co-abundance

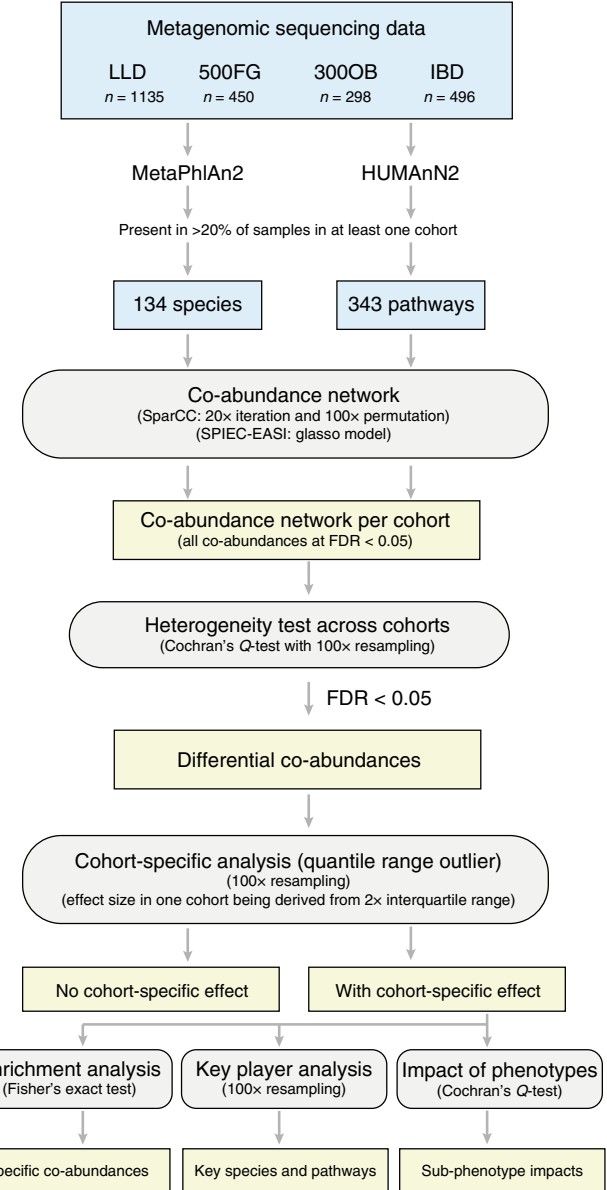

**Fig. 1 Analysis workflow of the present study.** The study comprised 2,379 metagenomics samples from four cohorts: the general population LifeLines-DEEP cohort (LLD), the 500FG cohort, an obesity cohort (300OB), and an inflammatory bowel disease (IBD) cohort. By combining SparCC and SPIEC-EASI methods, we constructed species and pathway co-abundance networks in each cohort separately. The established co-abundances were then subjected to heterogeneity testing and cohort-specificity analysis to assess whether the effect size of each co-abundance was significantly different between cohorts and whether the differences were driven by a specific cohort.

relationships by combining the SparCC[29] and SpiecEasi[38] methods. For species networks, we identified 2604 co-abundances in the LLD cohort, 1591 in the 500FG cohort, 1107 in the 300OB cohort and 2554 in the IBD cohort, yielding 3454 unique species co-abundances in total (false discovery rate (FDR) < 0.05, Fig. 2a, Supplementary Data 1). Notably, 82.1% of the species co-abundances also exhibited co-occurrence (Supplementary Fig. 3, Supplementary Data 1 and 2). For pathways, the numbers of co-abundances ranged from 37,279 in 500FG to 40,699 in LLD, yielding a total of 43,355 unique pathway co-abundances (FDR < 0.05, Fig. 2b, Supplementary Data 3). Since absence rate of bacterial pathways is much less than in bacterial

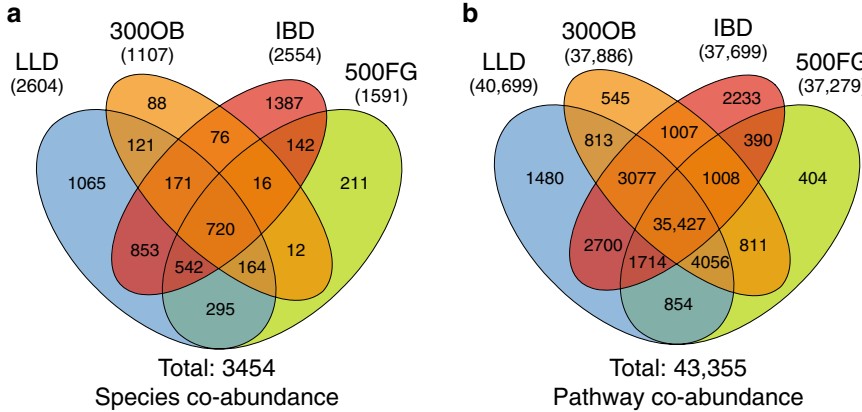

**Fig. 2 Microbial co-abundance networks in each cohort. a** Venn diagram of the numbers of species co-abundances detected in each cohort. In total, we identified 3454 co-abundance relationships significant at FDR < 0.05 in at least one cohort by combing SparCC and SpiecEasi. **b** Venn diagram of the numbers of species co-abundances detected in each cohort. Similarly, at the microbial metabolic pathway level, 43,355 co-abundance relationships were detected at FDR < 0.05 in at least one cohort by combing SparCC and SpiecEasi.

species, only 29.6% of pathway co-abundances showed co-occurrence (Supplementary Fig. 3, Supplementary Data 3 and 4). The co-occurrence results are summarized and further discussed in Supplementary Note 1.

**Microbial co-abundance strength varies between cohorts.** We hypothesized that co-abundance strengths could be different depending on host physiological status. We thus assessed to what extent the correlation coefficients were variable across cohorts and characterized variable co-abundance relationships for 38.6% of the species co-abundances and 64.3% of the pathway co-abundances (Cochran-Q test, FDR < 0.05, Supplementary Data 1 and 3).

**Differential microbial co-abundances are reflected in abundance levels.** When zooming in on the 100 species and 304 pathways that were involved in variable co-abundances, 76% of these species and 84% of these pathways also showed significant differences in their abundance levels among cohorts (analysis of variance test FDR < 0.05, Supplementary Data 5 and 6). This implies that the variable co-abundance relationship is largely reflected by differential microbial abundance. We summarized the number of differential co-abundances between species from the same genus or from different genera (Fig. 3a). The genus with the most heterogeneous co-abundances was *Streptococcus*, and a large number of variable co-abundances were observed not only between different *Streptococcus* species but also between *Streptococcus* species and species from other genera such as *Eubacterium* and *Veillonella* (Fig. 3a). In particular, *Streptococcus* species were higher in the IBD cohort, consistent with the results of previous studies[14,39]. A similar observation was found for the pathway co-abundances, particularly for amino acid biosynthesis pathways, which showed variability not only within themselves but also with respect to various pathways related to nucleoside and nucleotide biosynthesis (Fig. 3b).

**Specific microbial co-abundances are enriched in disease cohorts.** Next, we analysed whether the variable co-abundance relationships were driven by a particular cohort, i.e. whether the co-abundance strength in one cohort was very different from those in the other three cohorts. After correcting for the age and sex differences between cohorts, 120 species co-abundances (Supplementary Fig. 4) and 1448 pathway co-abundances (Supplementary Fig. 5) still showed cohort specificity with an FDR of 7.6%, as estimated by permutation (Supplementary Data 1 and 3). Interestingly, cohort-specific co-abundances were significantly enriched in the disease

cohorts compared to the population-based cohorts: 113 (94%) species co-abundances and 1050 (72%) pathway co-abundances were specifically related to the IBD cohort (Fisher's test $P = 1.2 \times 10^{-56}$ and $P < 10^{-260}$, respectively, Fig. 3c, d) and 281 (19.4%) pathway co-abundances were specifically related to the 300OB cohort (Fisher's test $P = 2.9 \times 10^{-29}$), as compared to only 3 species and 117 pathway co-abundance relationships specific to the population-based cohorts LLD and 500FG (Fig. 3c, d). Our results highlight that microbial co-abundances are dependent on host health and disease status. Below we discuss the microbial co-abundance networks in IBD and 300OB in more detail, further replicate our findings in independent cohorts and assess the relevance of disease subtypes, disease characteristics and medication usage.

**The microbial co-abundance network in IBD.** Replication of the IBD network in the integrative Human Microbiome Project (iHMP-IBD) cohort: Of the 2554 species and 37,699 pathway co-abundances established in our IBD cohort, we were able to assess 2090 species co-abundances and 37,106 pathway co-abundances in 77 IBD individuals from the iHMP-IBD[39]. In the baseline samples of the iHMP-IBD cohort, 531 species co-abundances (25.4%) and 21,882 (59.0%) pathway co-abundance could be replicated at $P < 0.05$ (Supplementary Data 7 and 8)[39]. The relatively low replication rate in species co-abundances is largely a power issue, as we also observed that 1705 (81.6%) species co-abundances and 24,165 (65.1%) pathway co-abundances showed no significant difference in their co-abundance strengths between our IBD cohort and the iHMP-IBD cohort (Cochran-Q test, $P > 0.05$, Supplementary Fig. 6, Supplementary Data 7 and 8). We then compared the IBD networks between the first and last time points of the iHMP-IBD cohort (~1 year apart) and replicated 90.6% of species co-abundances and 99.6% of pathway co-abundances (Cochran-Q test, $P > 0.05$, Supplementary Fig. 6, Supplementary Data 7 and 8). This suggests that our estimation of co-abundance strengths in IBD was largely replicable in a different cohort and was stable across time.

Microbial networks of IBD in relation to disease characteristics: Previous studies have shown that observed microbial abundance differences could be explained by certain disease characteristics of IBD[14]. We therefore hypothesized that this could also be the case for co-abundance relationships. We assessed whether IBD co-abundances (including IBD co-abundances at FDR < 0.05 and IBD-specific co-abundances) could be related to the disease subtypes [ulcerative colitis (UC, $n = 189$) vs. Crohn's disease (CD, $n = 276$)], disease location [ileum ($n = 212$) vs. colon ($n = 286$)] and disease activity

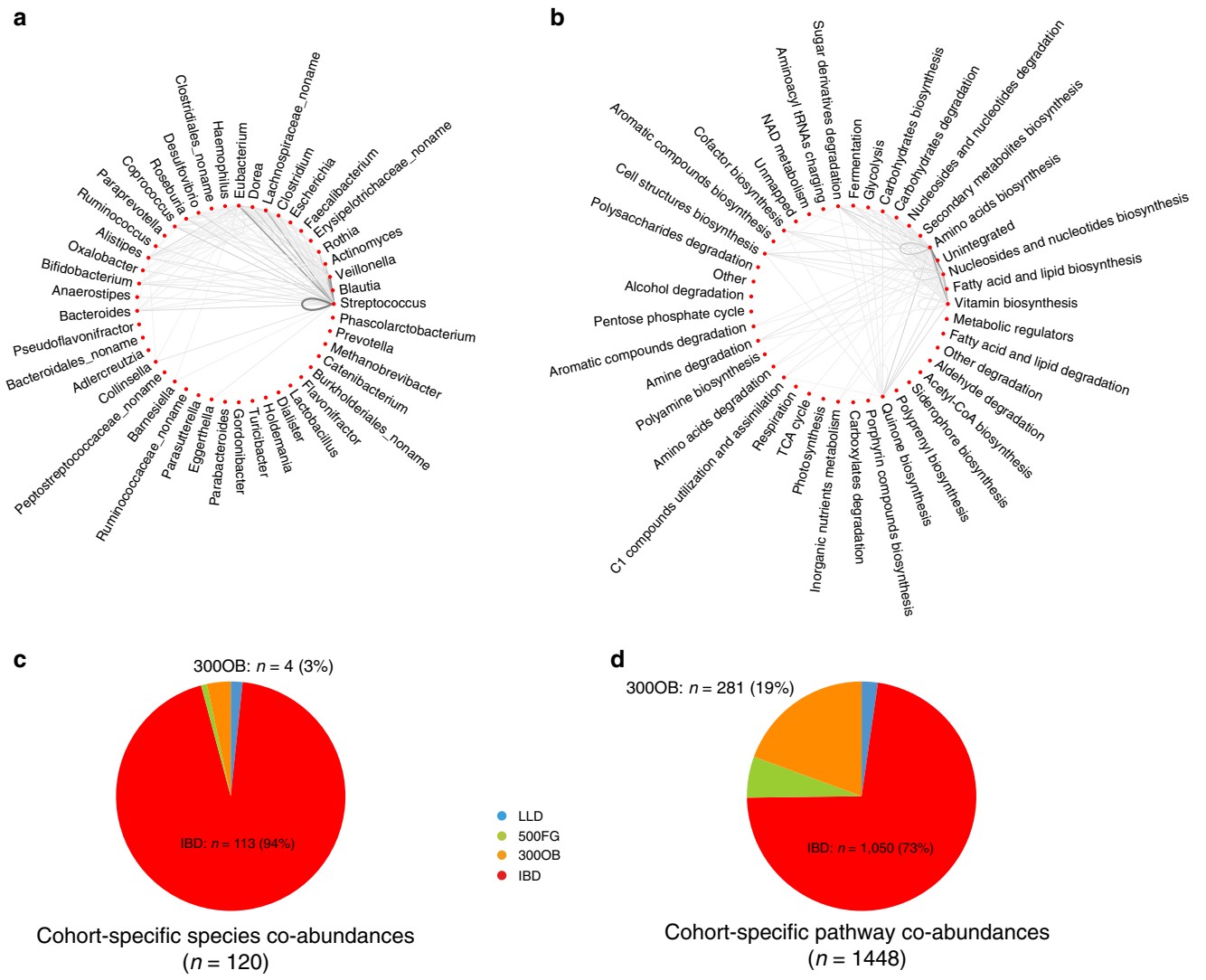

**Fig. 3 Differential and cohort-specific microbial co-abundances. a** Differential species co-abundances involved in 45 microbial genera. **b** Differential pathway co-abundances involved in 41 microbial metabolic categories. Each dot indicates one microbial genus or metabolic category. Each line represents differential species or pathway co-abundances between species or pathways from either the same or different genera or metabolic categories. The width and darkness of the lines represent the relative number of differential co-abundances. **c** Pie chart of 120 cohort-specific species co-abundances showing the proportion of specific co-abundances detected in each cohort. **d** Pie chart of 1448 cohort-specific pathway co-abundances showing the proportion of specific co-abundances detected in each cohort.

[inflammation ($n = 121$) vs. no inflammation ($n = 377$)] (Supplementary Table 1). Most of the co-abundance relationships were comparable between disease characteristics, and only a few showed significant differences at FDR < 0.05 (Supplementary Fig. 7, Supplementary Data 9 and 10), namely 16 species co-abundances related to disease subtypes and 8 species co-abundances related to location. For the pathway co-abundances, 91 were related to disease subtypes, 24 to location and 3 to activity (Cochran-$Q$ test FDR < 0.05, Supplementary Fig. 7). Out of these, five co-abundance relationships were related to an important butyrate producer, *Faecalibacterium prausnitzii*, which showed stronger co-abundance relationships in UC compared to CD. One example here was the negative co-abundance relationship of *F. prausnitzii* with *Haemophilus parainfluenzae*, a species known to have pathogenic properties[40].

Microbial networks of IBD in relation to medication: We further tested whether drug usage can affect microbial co-abundance, as usage of antibiotics (20.0%) and proton pump inhibitors (PPIs; 26.5%) was higher in patients with IBD than in the general population cohorts (1.1% and 8.4%) (Supplementary Table 1). Here

we detected no significant difference in species co-abundances between antibiotic users and non-users (Cochran-$Q$ test FDR > 0.05, Supplementary Fig. 7), while 1049 out of 37,959 (3.7%) pathway co-abundance relationships showed statistically significant differences between PPI users and non-users, in particular related to the isoprene biosynthesis and methylerythritol phosphate pathways (Cochran-$Q$ test FDR < 0.05, Supplementary Fig. 7, Supplementary Data 10).

Key species and pathways in IBD: When comparing microbial co-abundance in IBD to the other 3 cohorts, we identified 113 species co-abundances and 1050 pathway co-abundances that showed significantly different effects compared to the other 3 cohorts. We then assessed whether these IBD-specific co-abundances were highly connected to a specific pathway or species that may be disease relevant[24], and our analysis identified three key species and four key pathways for IBD (Fig. 4).

Key species included *Escherichia coli*, *Oxalobacter formigenes* and *Actinomyces graevenitzii*. *E. coli* and *O. formigenes* have previously been associated with IBD[14,41–45] (Fig. 4a, Supplementary Data 5).

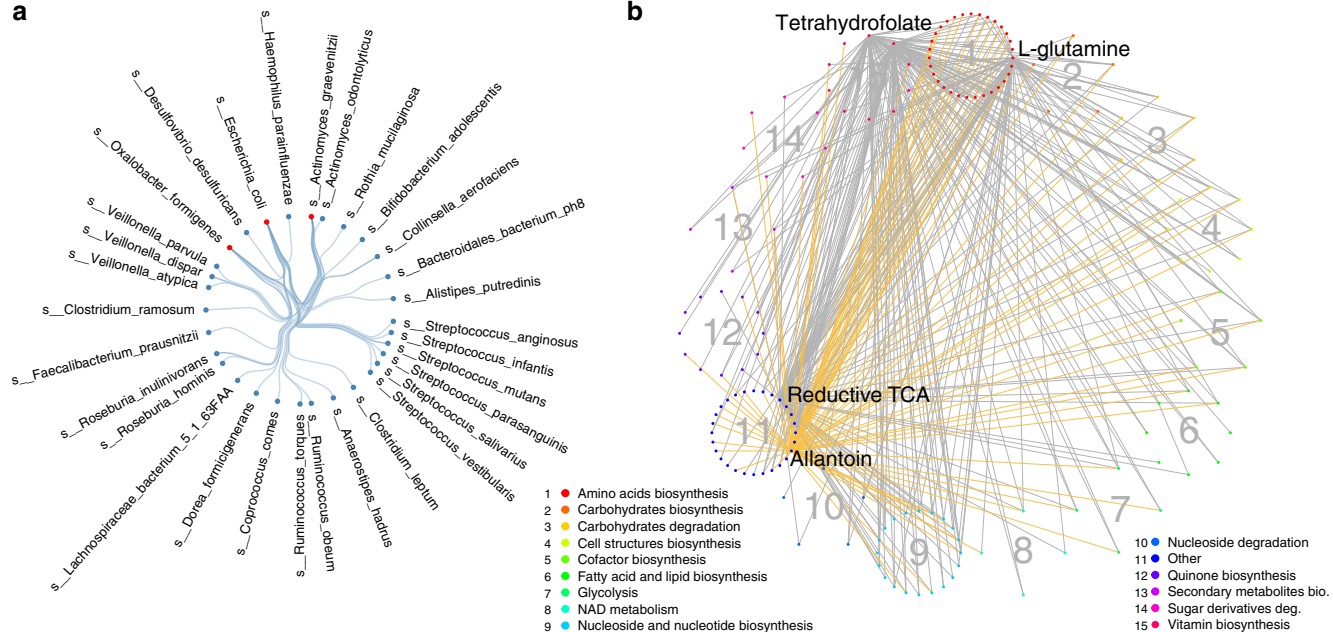

**Fig. 4 Cohort-specific species and pathway co-abundances. a** Cohort-specific co-abundances identified for three key species in the IBD cohort, involving 33 IBD-specific co-abundances. Each dot indicates one species. Red indicates IBD key species. Each line represents one IBD-specific co-abundance relationship. **b** Cohort-specific co-abundances identified for four key pathways in IBD and one key pathway in 300OB, involving 385 cohort-specific co-abundances. Each line represents a cohort-specific correlation between two pathways. Yellow lines represent obesity-specific co-abundances. Grey lines represent IBD-specific co-abundances. Each dot indicates one pathway. Pathways belonging to the same metabolic category have the same colour and are clustered as sub-circles. Colour legends are shown in the plot.

Interestingly, *E. coli* shows positive co-abundance relationships with species with pro-inflammatory properties, like *Streptococcus mutans*, and negative co-abundance relationships with species with anti-inflammatory properties, like *F. prausnitzii* (Supplementary Data 1). The one key species we identified for IBD, *A. graevenitzii*, is a microbe that is most often identified in the oral cavity or respiratory tract[46].

Key IBD pathways included a C1 compound utilization and assimilation pathway (P23-PWY: reductive tricarboxylic acid (TCA) cycle I), two vitamin biosynthesis pathways (FOLSYN-PWY: superpathway of tetrahydrofolate biosynthesis and salvage and PWY-6612: superpathway of tetrahydrofolate biosynthesis) and an amino acid biosynthesis pathway (PWY-5505: L-glutamate and L-glutamine biosynthesis) (Fig. 4b, Supplementary Data 6). The top key functional pathway in IBD was the reductive TCA cycle pathway (P23-PWY), which had 76 IBD-specific co-abundances, and 94.7% of these were replicated in the iHMP-IBD cohort (Supplementary Data 3 and 6). The reductive TCA cycle is a carbon dioxide fixation pathway that has been recognized as a primordial pathway for the production of organic molecules for the biosynthesis of sugars, lipids, amino acids, pyrimidines and menaquinone (Fig. 5a)[47]. For instance, one IBD-specific co-abundance relationship was related to the biosynthesis of menaquinone (PWY-5837), which is also known as vitamin K2. The co-abundance relationship for this pathway in IBD ($r = 0.1$) was weaker than in other cohorts ($r = 0.3$) (Fig. 5b), despite the higher abundance of this pathway in IBD (FDR < 0.05, Fig. 5c, Supplementary Data 6). *E. coli* is known to be an important species for the biosynthesis of menaquinone, a growth-promoting factor for a variety of microorganisms in the gut microbiota[48]. In line with this, we found that 18.8% of the menaquinone biosynthesis pathway in IBD patients was contributed by *E. coli*, two times higher than in the two population-based cohorts (Wilcoxon test $P < 3.0 \times 10^{-11}$; Supplementary Data 11). This finding suggests *E. coli* as an important contributor to menaquinone

biosynthesis in IBD that may promote the growth of other microorganisms. Indeed, our study also revealed *E. coli* as a key IBD species, exerting IBD-specific co-abundance relationships with 15 species (Supplementary Data 5). Of these, strong positive correlation was observed for inflammation-related *Streptococcus* species, including *S. mutans*[49], *Streptococcus vestibularis*[41] and *Streptococcus infantis*[42] (Fig. 5d). Accordingly, higher correlations were observed between menaquinone biosynthesis and *Streptococcus* species in IBD than in the other cohorts (Fig. 5e, Supplementary Data 12).

**The microbial co-abundance network in 300OB.** Replication of 300OB network in LLD obese individuals: 1107 species and 37,886 pathway co-abundances were detected in the 300OB cohort (Fig. 2). These estimated co-abundance strengths were largely replicable in 134 obese individuals with matched age and body mass index (BMI) from the LLD cohort, with 991 (89.5%) species co-abundances and 32,963 (87.0%) pathway co-abundances showing no difference (Cochran-Q test $P > 0.05$, Supplementary Fig. 8, Supplementary Data 13 and 14).

Microbial networks in relation to obesity-related diseases: The 300OB cohort was set up to study cardiovascular disease in obese individuals, including 139 patients with atherosclerotic plaque and 159 obese controls (Supplementary Table 1). In addition, 35 300OB participants had diabetes. Here we observed only three species co-abundances related to cardiovascular disease, with all 3 showing stronger co-abundances in patients with plaque than in patients without (Cochran-Q test FDR < 0.05, Supplementary Fig. 9, Supplementary Data 13 and 14). These were positive co-abundances between *Dorea longicatena* and *Dorea formicigener-ans* and negative co-abundances of *Lachnospiraceae bacterium 9.1.43BFAA* with *Coprococcus comes* and *D. longicatena*.

Key pathways in obesity: When we compared microbial co-abundances in the 300OB to the other 3 cohorts, we identified 281

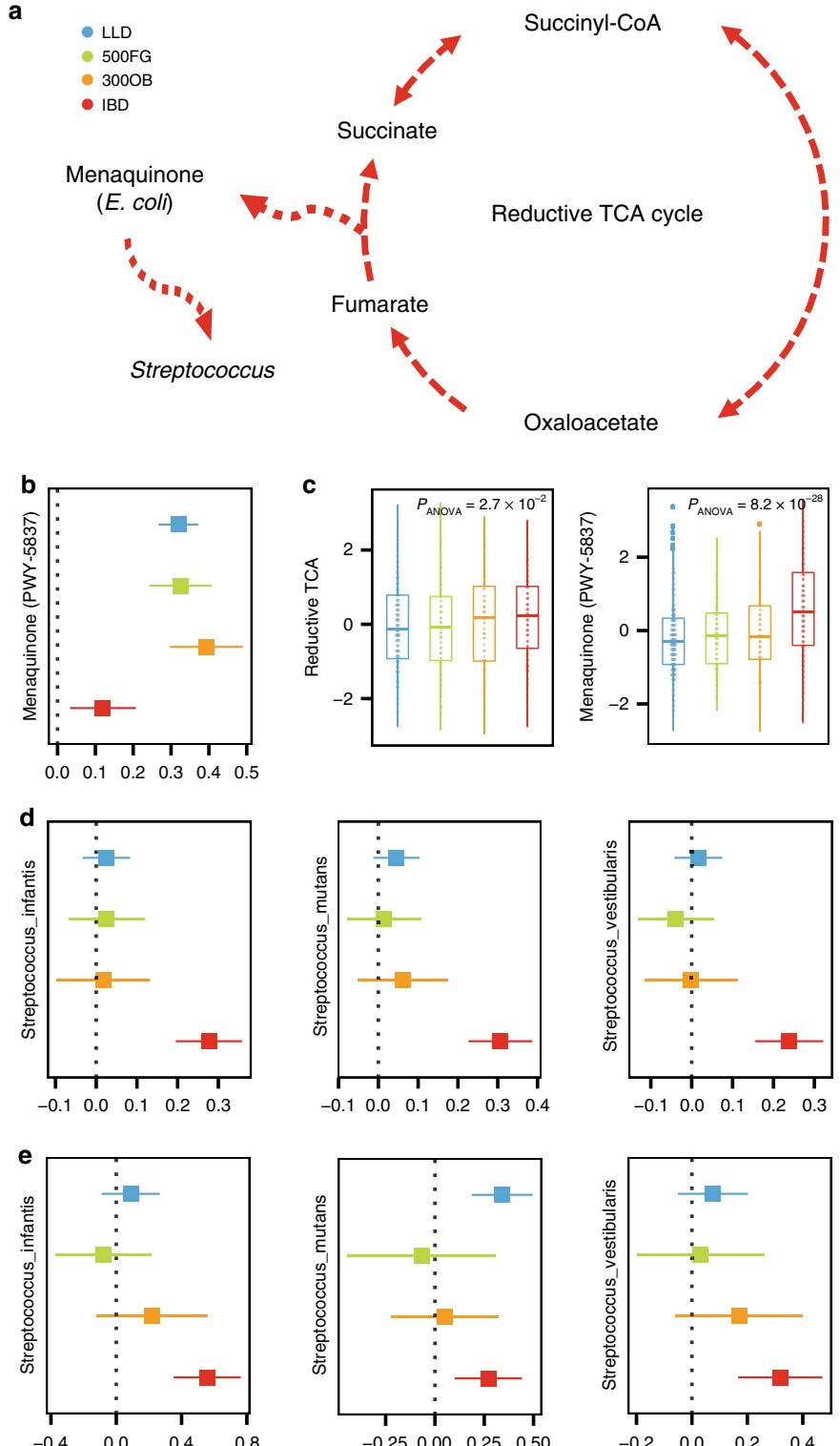

**Fig. 5 Menaquinone biosynthesis related to Streptococcus overgrowth in IBD. a** Menaquinone biosynthesis (PWY-5837) from the reductive TCA cycle (P23-PWY) in bacteria. **b** The menaquinone biosynthesis pathway shows IBD-specific interaction with the reductive TCA cycle pathway. **c** Both menaquinone biosynthesis and reductive TCA cycle pathway abundance are significantly higher (ANOVA test, FDR < 0.05) in the IBD cohort than in the two population-based cohorts. Box plots show medians and the first and third quartiles (the 25th and 75th percentiles) of abundance after correcting for age and sex, respectively. The upper and lower whiskers extend the largest and smallest value no further than 1.5 × IQR, respectively. Outliers are plotted individually. (Source data is provided as a Source data file). **d** Three *Streptococcus* species show IBD-specific co-abundance with *Escherichia coli*. **e** The menaquinone biosynthesis pathway shows strong positive correlation with three *Streptococcus* species in IBD. $N = 2379$ independent samples are involved ($N_{LLD} = 1135$, $N_{500FG} = 450$, $N_{300OB} = 298$, $N_{IBD} = 496$). The forest plots show co-abundance strength and direction in each cohort, with square dot for the correlation coefficient and bar for the 95% confidence interval.

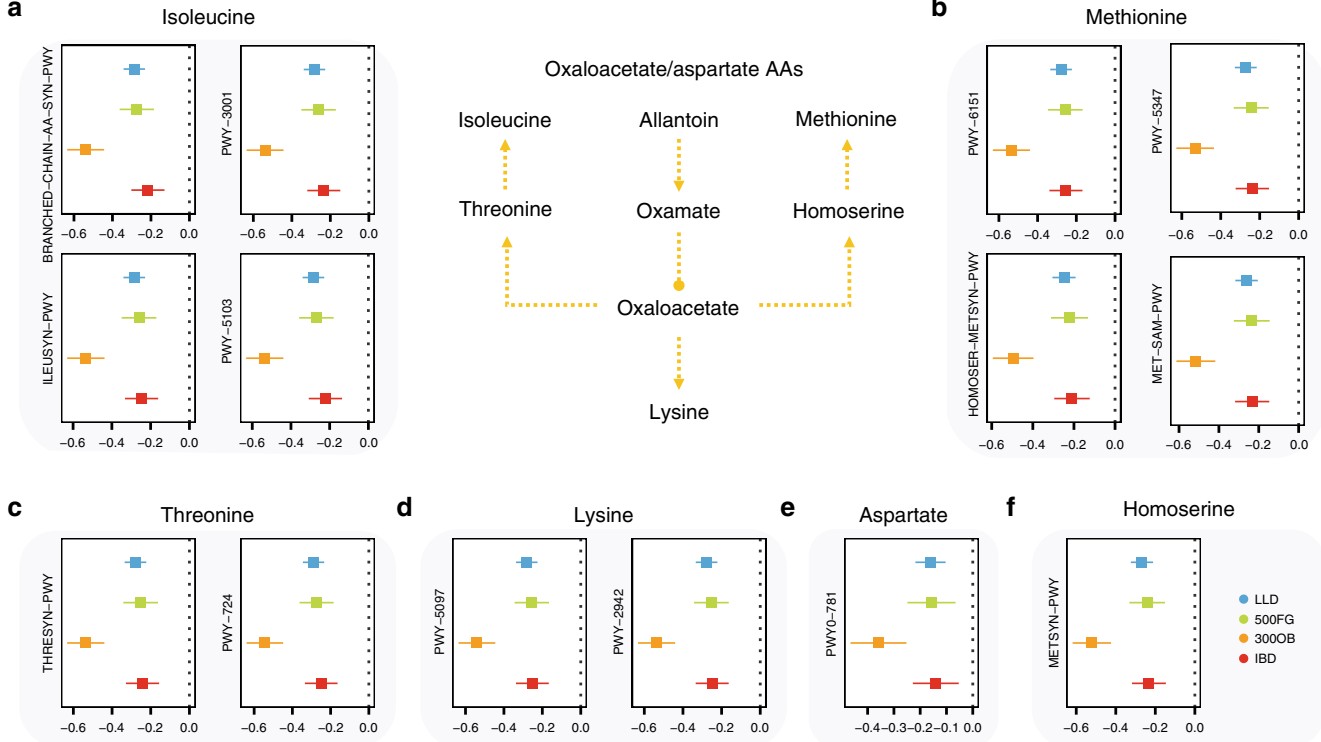

**Fig. 6 Allantoin degradation pathway links to glycaemia in obesity.** The allantoin degradation pathway shows stronger negative correlation with 14 amino acid biosynthesis pathways in the obesity cohort compared to the other cohorts. These pathways represent the biosynthesis of six amino acids: **a** isoleucine (PWY-5103, PWY-3001, BRANCHED-CHAIN-AA-SYN-PWY and ILEUSYN-PWY), **b** methionine (PWY-6151, PWY-5347, MET-SAM-PWY and HOMOSER-METSYN-PWY), **c** threonine (THRESYN-PWY and PWY-724), **d** lysine (PWY-5097 and PWY-2942), **e** aspartate (PWY0-781) and **f** homoserine (METSYN-PWY). All six amino acids are involved in the oxaloacetate/aspartate amino acids biosynthesis pathway. Lines with arrows represent metabolic relationships. Lines with a circle represent an inhibitory role in a metabolic pathway. $N = 2379$ independent samples are involved ($N_{LLD} = 1135$, $N_{500FG} = 450$, $N_{300OB} = 298$, $N_{IBD} = 496$). The forest plots show co-abundance strength and direction in each cohort, with square dot for the correlation coefficient and bar for the 95% confidence interval.

pathway co-abundances that showed a significantly different effect, i.e. obesity-specific co-abundances. One key pathway in obesity was degradation of allantoin (PWY0-41, Fig. 4b, Supplementary Data 6), which showed obesity-specific co-abundance relationships with 85 pathways. Allantoin is one of the active principles in various plants, e.g. yams, and is found to enhance insulin secretion and lower plasma glucose[43,44]. Its degradation product, oxamate, plays an inhibitory role in oxaloacetate/aspartate amino acids[45]. In line with this, we found that the allantoin degradation pathway showed stronger negative correlations with the biosynthesis pathways of oxaloacetate/aspartate amino acids (including lysine, homoserine, methionine, threonine and isoleucine) and the biosynthesis pathway of aspartate (PWY0-781, Fig. 6), which were both positively associated with fasting glucose level and negatively associated with fasting insulin level ($P < 0.05$, Supplementary Table 2).

## Discussion
This study is a microbial co-abundance network analysis based on metagenomics data, involving 2379 participants from two population-based cohorts (LLD and 500FG) and two disease cohorts (IBD and 300OB). We report 3454 species and 43,355 pathway co-abundance relationships that were significant in at least one cohort. Among them, the effect sizes of 38.6% of species co-abundances and 64.3% of pathway co-abundances were significantly different between cohorts. In particular, 113 species co-abundances and 1050 pathway co-abundances showed IBD cohort-specific effects and 281 pathway co-abundances had specific effects in the 300OB cohort. Our study provides evidence

that microbial dysbiosis can be reflected in alterations in microbial co-abundance.

Our study yielded several findings. We identified three species and four pathways in IBD and one pathway in 300OB that served as key players in disease-specific co-abundance networks. Key IBD-associated species included *E. coli* and *O. formigenes*[14,50,51]. A higher abundance of the pathogenic species *E. coli* has previously been associated with IBD[50,51], likely due to an increased release of oxidized haemoglobin into the intestinal lumen as a result of chronic inflammation of the gastrointestinal walls[52,53]. Consistent with this, we replicated high abundances of *E. coli* and low abundances of anaerobic metabolism pathways in IBD. *E. coli* also showed strong positive co-abundance with other inflammation-inducing species in IBD, including streptococcus species such as *S. mutans*[49], *S. vestibularis*[41] and *S. infantis*[42]. In contrast, these co-abundances were either weak or negative in our population-based and obesity cohorts. We further identified *A. graevenitzii* as a key species in IBD. Although no evidence supports a direct role for *Actinomyces* in the pathogenesis of IBD, *A. graevenitzii* has been associated with coeliac disease in children[54] and can induce actinomycosis[55,56], with both conditions sharing similar abdominal pathologies with IBD[57]. Two case reports have also suggested that *Actinomyces* may aggravate the intestinal injuries caused by inflammation[58,59].

The top key functional pathway in IBD was the reductive TCA cycle pathway (P23-PWY), which had 76 IBD-specific co-abundances. Interestingly, the key IBD species *E. coli* is known to be an important species for the biosynthesis of menaquinone, a growth-promoting factor for a variety of microorganisms in the gut

microbiota[48]. In line with this, we found that 18.8% of the menaquinone biosynthesis pathway in IBD patients was attributed to *E. coli*, which is two times higher than in the two population-based cohorts (Wilcoxon test $P < 3.0 \times 10^{-11}$). Another notable IBD key pathway is the tetrahydrofolate pathway, which is responsible for folic acid derivative biosynthesis and supplementation of folic acid. This pathway has been shown to reduce the risk of colorectal cancer in IBD patients[60]. Interestingly, previous research has shown that oral intake of L-glutamine attenuates the colitis induced by dextran sulfate sodium in mice[61]. We identified a negative co-abundance with L-glutamine biosynthesis and biosynthesis of other amino acids like L-isoleucine and L-methionine. Previous research showed that both these amino acids play an important role in the immune system[62,63]. L-glutamine has been tested as supplement in patients with IBD but did not show improvements in clinical outcomes like disease activity scores[64]. Our results show large numbers of connections for L-glutamine with other pathways such as the biosynthesis of other amino acids. These pathways might also be of interest when exploring L-glutamine as an intervention for IBD.

In obesity, we identified the allantoin degradation pathway as a key pathway, showing obesity-specific co-abundance relationship to 85 pathways, mainly negative correlations with biosynthesis of oxaloacetate/aspartate amino acids. These pathways are related to insulin secretion and glucose metabolism. However, their co-abundance relationships did not show significant differences between patients and non-diabetic individuals, which is likely due to a power issue as there were only 35 diabetic patients in 300OB. Instead, we found three species co-abundances related to presence of atheriosclerotic plaque, involving *D. longicatena*, *D. formicigenerans*, *L. bacterium 9.1.43BFAA* and *C. comes*. Notably, *D. Longticatena* and *Lachnospiraceae* species have been linked to atherosclerotic cardiovascular disease[9].

Altogether, our analyses show that microbial dysbiosis in disease may not be driven solely by differences in abundance level, it may also reflect shifts in microbial interactions that are mirrored in co-abundance analyses. Particularly when applied to metagenomics sequence data, pathway-based co-abundance networks provide further insights into functional dysbiosis in IBD and obesity. However, we also acknowledge several limitations of our study. This is an in silico network analysis based on correlation in bacterial abundance levels. Even with the large sample size, our study is still undersized for making comparisons to the number of interactions assessed. In recent years, many different network tools have been developed to tackle the statistical challenges in inferring networks for compositional data. In this study, we applied two independent methods, SparCC and SpiecEasi, to establish microbial co-abundance networks based on MetaPhlan and HUMAnN2 annotation. Our analysis can thus be biased due to these annotation tools. Other annotation tools, e.g. mcSEED[65], may yield different pictures of microbial community and functional profile, thereby identifying different co-abundance networks. Thus such in silico-based network inferences require further functional validation. Although bacterial genes are believed to be expressed uniformly[66], previous studies have also shown that meta-transcription can exert dynamic changes in response to environmental perturbations that cannot be detected at the metagenome level[67,68]. Thus, in order to understand the microbial ecosystem in terms of functional interaction in diseases, we need complementary approaches like meta-proteomics and meta-metabolomics that provide a more direct readout of the functional properties of the gut microbiome. Furthermore, the cross-sectional design of this study makes it hard to assess the stability of our findings over time. However, we did observe similar findings for the iHMP-IBD cohort for 98.2% of species co-abundances and 99.4% of pathway co-abundances between two time points spanning 1 year (Cochran-*Q* test $P > 0.05$).

This implies that co-abundance relationships are largely consistent over time.

In addition, due to our study design, we cannot disentangle cause from consequence. Longitudinal studies are therefore warranted and should be combined with functional validation. Moreover, especially in the context of IBD, which is a heterogeneous disease, we had limited ability to pinpoint co-abundance networks to specific disease characteristics like the subtypes CD and UC. This is probably due to the lack of power to detect this by subgrouping our cohorts. Larger cohorts with well-documented disease characteristics are needed in the future.

This study presents the microbial network analysis to examine both microbial species and functional pathways based on metagenomics sequencing. Our data show that dysbiosis of the gut microbial ecosystem in disease can not only be assessed by the altered abundance level but can also be seen at the level of microbial interaction, at least in terms of co-abundances. We have also identified IBD- and obesity-specific species and pathways that potentially play important roles in regulating the microbial ecosystem in disease, and these disease-specific microbial interactions extend our current knowledge about the role of the microbiome in disease.

## Methods

**Study cohorts**. All four cohorts used in this study have been described before[3,14,69,70]. In short, the LLD cohort is a large prospective cohort study from the north of the Netherlands[71]. LLD contains 58.20% females and 41.80% males, the mean age (SD) of participants is 45.04 (13.60) years and their mean BMI is 25.26 (4.18) (Supplementary Fig. 1). In this study, we included 1135 LLD individuals for whom there is metagenomics and phenotype data[3].

The 500FG cohort consists of 534 healthy adult volunteers from the Netherlands[69,70]. In 500FG, 56.50% are women and 43.50% are men, the mean age of participants is 27.43 (12.35) years and their mean BMI is 22.70 (2.72) (Supplementary Fig. 1). In this study, we included 450 500FG participants for whom metagenomics data are available.

The 300OB is a part of the Functional Genomics project and consists of 302 individuals from the Netherlands with a BMI >27[70,72,73]. These individuals have been clinically screened for obesity-related comorbidities. Around half of participants are clinically diagnosed with metabolic syndrome. 300OB is 55.70% male and 44.30% female, the mean age of participants is 67.07 (5.39) years and their mean BMI is 30.73 (3.48) (Supplementary Fig. 1). In this study, we included 298 participants from 300OB for whom metagenomics data are available.

The 1000 Inflammatory Bowel Disease (1000IBD) cohort consists of patients with IBD recruited at the specialized IBD outpatient clinic of the University Medical Center Groningen in the Netherlands[14,74]. IBD diagnosis was made based on accepted radiological, endoscopic and histopathological evaluation. The 1000IBD cohort is 60.70% female and 39.30% male, the mean age of participants is 43.45 (14.52) years and their mean BMI is 25.55 (5.17) (Supplementary Fig. 1). In this study, we included 496 IBD participants for whom metagenomics data are available.

**Ethical approval**. All participants signed an informed consent form prior to sample collection. Institutional ethics review board (IRB) approval was available for all four cohorts: the LLD (ref. M12.113965) and the IBD (IRB-number 2008.338) cohorts were approved by the UMCG IRB and the 500FG study (NL42561.091.12, 2012/550) and 300OB (NL46846.091.13) cohorts were approved by the Ethical Committee of Radboud University Nijmegen.

**Metagenomic data generation and pre-processing**. All participants from the four cohorts were asked to collect faecal samples at home and to place them in their home freezer (−20 °C) within 15 min after production. Subsequently, a nurse visited the participant to pick up the faecal samples on dry ice and transfer them to the laboratory. Aliquots were then made and stored at −80 °C until further processing. The same protocol for faecal DNA isolation and metagenomics sequencing was used for all four cohorts. Faecal DNA isolation was performed using the AllPrep DNA/RNA Mini Kit (Qiagen, cat. 80204). After DNA extraction, faecal DNA was sent to the Broad Institute of Harvard and MIT in Cambridge, MA, USA, where library preparation and whole-genome shotgun sequencing were performed on the Illumina HiSeq platform. From the raw metagenomic sequencing data, low-quality reads were discarded by the sequencing facility and reads belonging to the human genome were removed by mapping the data to the human reference genome (version NCBI37) with Bowtie2 (v2.1.0)[75].

The relative abundance of gut microbial taxonomic units was determined using MetaPhlan (v2.7.2)[76], and the relative abundances of metabolic pathways were determined using the HUMAnN2 pipeline (v0.10.0)[77], which maps DNA/RNA

reads to a customized database of functionally annotated pan-genomes. HUMAnN2 reported the abundances of gene families from the UniProt Reference Clusters[78] (UniRef90), which were further mapped to microbial pathways from the MetaCyc metabolic pathway database[79,80]. In total, we detected 698 species and 489 pathways present in at least 1 of the 4 cohorts. To deal with sparse microbial data in the network analysis, we focused on species/pathways present in at least 20% of samples in at least one cohort. This provided a confined list of 134 species and 343 pathways for use in the network analysis. Together these accounted for, on average, 86.9% and 99.9% of taxonomic and functional compositions, respectively.

**Statistical analysis**. Co-abundance network inference: co-abundance analysis on compositional data is challenging because it is likely to exhibit spurious correlations due to the dependency of fractions (i.e. relative abundance sums to 1)[29,81–84]. In particular, the problem can be more serious in a microbial community with low compositionality[85]. We therefore first assessed the inverse Simpson index of microbial composition for the effective number of species ($n_{eff}$). Our analysis showed high compositionality in both functional pathway composition (2.09, 2.10, 2.11 and 2.08 in LLD, 500FG, 300OB and IBD, respectively) and species composition (10.74, 11.87, 12.30 and 8.80 in LLD, 500FG, 300OB and IBD, respectively). Following the suggestion of Weiss et al.[85], based on their assessment of the performance of eight different methods (Bray–Curtis, Pearson, Spearman, CoNet, LSA, MIC, RMT and SparCC), we decided to use the SparCC method because it has been proven to be able to infer linear relationships with high precision for high diversity compositions with $n_{eff} < 13$. Species composition data from MetaPhlan was converted to predicted read counts by multiplying relative abundances by the total sequence counts and then subjected to a Python-based SparCC tool[29]. For pathway analysis, the read counts from HUMAnN2 were directly used for SparCC. Significant co-abundance was controlled at FDR 0.05 level using 100× permutation. In each permutation, the abundance of each microbial factor was randomly shuffled across samples. To reduce indirect associations, we further applied SpiecEasi (v1.0.6), which infers the microbial network underlying graphical model using the concept of conditional independence[38]. In this way, we obtained 3454 species and 43,355 pathway co-abundances that were detectable by both methods (Fig. 1).

Co-occurrence network inference: presence and absence of each bacterial species and metabolic pathway were treated as binary traits. The pair-wise co-occurrence relationship between two microbial factors (species or pathway) in each cohort was assessed using Pearson's chi-squared test. If the number of co-occurrence pairs was greater than the number of co-exclusion pairs, the two microbial factors were considered to be a co-occurrence. If the number of co-occurrence pairs was less than the number co-exclusion pairs, the two factors were considered to be a co-exclusion. Permutation (100×) was conducted to determine significance at an FDR < 0.05. In each permutation, the presence and absence of each microbial factor was randomly shuffled across samples. At the species level, we detected 6015 co-occurrence relationships that were found in at least one cohort, with 3423 found in LLD, 1845 in 500FG, 1199 in 300OB and 4701 in IBD (Supplementary Data 2). At the pathway level, we detected 19,903 co-occurrence relationships that appeared in at least one cohort, with 13,501 found in LLD, 7581 in 500FG, 7580 in 300OB and 16,596 in IBD (Supplementary Data 4).

Network heterogeneity analysis: To assess the variability of networks among the four cohorts, we conducted Cochran-Q tests to assess the heterogeneity of effect sizes and directions across the four cohorts for each co-abundance (correlation coefficient generated by SparCC) and co-occurrence (odds ratio). Here we treated each cohort as one study and conducted the Cochran-Q test using the *metagen()* function from the package meta (v4.9.5) in R, which calculates the squared difference between individual study effects and the pooled effect using inverse variance weighting[86]. For each co-abundance, the P values from the Cochran-Q test were recorded, and co-abundances with significant heterogeneity were controlled at the FDR 0.05 level determined by permutation (100×). In this case, all samples from the four cohorts were randomly shuffled across cohorts, i.e. shuffling the cohort labels but keeping the correlation structure of species and pathways intact. Co-abundances with a Cochran-Q test FDR < 0.05 were considered heterogeneous, while co-abundances with Cochran-Q test P > 0.05 were considered stable. We also summarized species co-abundance co-abundances based on microbial genus and pathway co-abundance co-abundances based on metabolic category.

**Cohort-specific co-abundance selection**. For heterogeneous co-abundances and co-occurrences (Cochran-Q test FDR < 0.05), we further assessed whether these relationships showed cohort specificity, i.e. whether the effect size of co-abundance/co-occurrence in one cohort was very different from that in the other three. In this analysis, effect size for co-abundance was the SparCC correlation coefficient and the odds ratio for co-occurrence. We adopted interquartile ranges (IQRs) based the outlier detection method (Supplementary Fig. 10)[87]. We ranked the effect sizes from low to high, say b1, b2, b3, b4, and then calculated the corresponding 25%, 50% and 75% quartile values (Q1, Q2 and Q3, respectively). IQR was then calculated, and we assessed whether the smallest or largest effect size fell outside of $Q1 − 2 × IQR$ or $Q3 + 2 × IQR$. If only one met the condition, we called this co-abundance specific and assigned it to the corresponding cohort (Supplementary Fig. 10). To assess whether cohort-specific co-abundances were enriched for a specific cohort, we conducted Fisher's exact test. We also calculated the average

FDR of cohort-specific co-abundances using 100× permutations as described above for the heterogeneity analysis.

**Key microbial species and pathway detection**. To assess to what extent cohort-specific microbial relationships were linked to a specific species or pathway, we calculated the number of cohort-specific microbial relationships per species/pathway. To define the key species/pathway, we took the maximum number of false cohort-specific relationships per species/pathway from each permutation and determined the key species/pathway cut-off as the upper range of the 95% of confidence interval based on 100× permutations. At this cut-off, there is a 5% probability that a false enrichment could occur by chance. In this way, a species with at least 13 cohort-specific co-abundances or a pathway with at least 70 cohort-specific co-abundances was recognized as a key species or pathway. For co-occurrence networks, these numbers were 10 for key species and 45 for key pathways. In such a way, we detected 192 cohort-specific species co-abundances and 2235 cohort-specific pathway co-abundances.

**Assessing impact of confounding factors**. The age and sex distributions were different between cohorts (Supplementary Fig. 1). To assess the impact of age and sex, we conducted partial correlation analysis (Supplementary Fig. 11). For example, to assess the co-abundance between species A and B, we first assessed the Pearson correlation of A and B to each covariant, say C, respectively. Then, a pairwise correlation matrix of A, B and C was subjected to partial correlation (Supplementary Fig. 11) using the partial correlation function *cor2pcor* from the R package corpcor (version 1.6.9). This insured that the partial correlation determined between A and B was independent of the covariant C. To assess the impact, we compared the correlation coefficient between SparCC correlation and partial correlation for all co-abundances and found comparable effect size (Supplementary Fig. 12). After regressing out the confounding effects of age and sex on cohort-specific co-abundances, 120 out of 192 (62.5%) species and 1448 out of 2235 (64.8%) pathway co-abundances remained cohort specific.

**Replication of microbial networks**. To replicate microbial networks in IBD, we used data from 77 IBD patients from the iHMP-IBD as a replication cohort[88]. Given the iHMP-IBD's longitudinal study design, we could examine metagenomics data from the first and the last sample collection for each individual. In all, 91% of the species (123 out of 134) and 99% of the pathways (340 out of 343) found in our IBD cohort were also detected in the first sample collection in iHMP-IBD. The differences in co-abundance strength between the IBD cohort and the iHMP-IBD cohort were assessed using Cochran-Q test. A significant P > 0.05 was applied to define replicable co-abundances. We also investigated the stability of microbial networks in iHMP-IBD by comparing the microbial co-abundances in the first and last sample collection from the same participants using the same approach.

To replicate microbial networks in 300OB, we selected 134 obese individuals from the LLD cohort with matched age and BMI. Here we considered a co-abundance to be replicable if the Cochran-Q test heterogeneity between the discovery and replication cohorts was not significant at P > 0.05.

**Assessing the relevance of microbial co-abundances to sub-phenotypes**. Patients in the IBD and 300OB cohorts have different disease subtypes, and both cohorts had higher proportions of drug users than our population cohorts. In particular, the IBD cohort contained 276 patients with CD and 189 with UC. Within the IBD cohort, 126 patients took PPIs and 97 took antibiotics. In the 300OB cohort, 53.4% (159 out of 298) had an atherosclerotic plaque detected by ultrasound[72] and 35 were diabetic. To assess the co-abundance related to disease sub-phenotypes, we split the cohorts based on disease subtypes or medication use and inferred microbial co-abundance using SparCC. Cochran-Q test was applied to assess the differential microbial co-abundances at FDR < 0.05.

**Species contributions to pathways and species–pathway associations**. Since the pathway abundances reported by HUMAnN2 are computed at both community and individual species level[77], we further looked into the contribution of species to each pathway and reported the top contributor (species). To show the functional relationship between species and pathways (e.g. whether a given pathway has the potential to promote the growth of a species through its metabolic products), we also checked the correlation (Spearman) between microbial species and pathway abundance after adjusting for age, sex and read depth using a linear regression model[89]. FDR was further calculated based on 100× permutation.

**Network visualization**. Cohort-specific networks based on cohort-specific co-abundances were visualized using a circle plot or heatmap with hierarchical clustering analysis (ward.D clustering based on Minkowski distance). Both species and pathways networks were visualized using the package igraph (v1.2.4.1)[90] in R. For species networks, species belonging to the same genus were clustered together. For pathway networks, pathways from the same metabolic category were presented in a sub-circle, and categories with a limited number of pathways (<4) were grouped into the other category. Classification of pathways was based on the MetaCyc metabolic pathway database[79,80].

**Reporting summary**. Further information on experimental design is available in the Nature Research Reporting Summary linked to this paper.

## Data availability

All relevant data supporting the key findings of this study are available within the article and its Supplementary Information files. Data underlying Fig. 5c and Supplementary Fig. 2 are provided as a Source data file. Data underlying all the other figures are provided in Supplementary Data and data repositories: LifeLines-Deep cohort [https://www.ebi.ac.uk/ega/datasets/EGAD00001001991], 1000 IBD cohort [https://www.ebi.ac.uk/ega/datasets/EGAD00001004194], 300OB cohort [https://ega-archive.org/dacs/EGAC00001001143], and 500FG cohort [https://www.ncbi.nlm.nih.gov/bioproject/?term=PRJNA319574]. The iHMP data are available via https://ibdmdb.org/tunnel/public/summary.html. Due to informed consent regulation, the data sets of the Lifelines-DEEP, IBD, 300OB and 500FG cohorts are available upon request to the University Medical Center of Groningen (UMCG), Lifelines and Radboud University Medical Center, respectively. This includes the submission of a letter of intention to the corresponding data access committee [the Lifelines Data Access Committee for the LifeLines-DEEP data (Jackie Dekens, e-mail: j.a.m.dekens@umcg.nl), 1000 IBD Data access Committee UMCG for the IBD data (Melinde E. Wijers, e-mail: m.e.wijers@umcg.nl) and the Human Functional Genomics Data Access Committee for 500FG and 300OB data (Martin Jaeger, e-mail: Martin.Jaeger@radboudumc.nl)]. Data sets can be made available under a data transfer agreement and the data usage access is subject to local rules and regulations. Source data are provided with this paper.

## Code availability

For this study, the following software was used: kneadData (v0.4.6.1), Bowtie2 (v2.1.0), MetaPhlAn2 (v2.7.2), HUMAnN2 (v0.10.0), SparCC Python package, R (v3.5.2), SpiecEasi R package (v1.0.6), and meta R package (v4.9.5). Code used for generating the microbial abundance profiles is publicly available at https://github.com/GRONINGEN-MICROBIOME-CENTRE/Groningen-Microbiome/blob/master/Scripts/Metagenomics_pipeline_v1.md. Code used for the statistical analyses is publicly available at https://github.com/GRONINGEN-MICROBIOME-CENTRE/Groningen-Microbiome/tree/master/Projects/Microbial%20co-abundance%20network. Source data are provided with this paper.

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

## Acknowledgements

We thank the participants and staff of LifeLines-DEEP, 500FG, 300OB and the IBD cohort for their collaboration and the support of the LifeLines Cohort study and the Human Functional Genomics Project. We thank J. Dekens, M. Platteel, A. Maatman and J. Arends for management and technical support and K. Mc Intyre for English editing. This project was funded by the Netherlands Heart Foundation (IN-CONTROL CVON grant 2012-03 and 2018-27 to L.A.B.J., N.P.R., M.G.N., F.K., A.Z. and J.F.); the Top Institute Food and Nutrition, Wageningen, the Netherlands (TiFN GH001 to C.W.); the Netherlands Organization for Scientific Research (NWO) (NWO-VIDI 864.13.013 to J.F., NWO-VIDI 016.178.056 to A.Z., NWO Spinoza Prize SPI 94-212 to M.G.N., NWO Spinoza Prize SPI 92-266 to C.W. and NWO Gravitation Netherlands Organ-on-Chip Initiative (024.003.001) to C.W.); the European Research Council (ERC) (FP7/2007-2013/ERC Advanced Grant Agreement 2012-322698 to C.W., ERC Consolidator Grant 310372 to M.G.N. and ERC Starting Grant 715772 to A.Z.); the Stiftelsen Kristian Gerhard Jebsen Foundation (Norway) to C.W.; L.A.B.J. was supported by a Competitiveness Operational Programme grant of the Romanian Ministry of European Funds (P_37_762, MySMIS 103587); the RuG Investment Agenda Grant Personalized Health to C.W. and the Foundation De Cock-Hadders grant (20:20-13) to L.C. A.Z. holds a Rosalind Franklin Fellowship from the University of Groningen. L.C. is supported by a joint fellowship from the University Medical Center Groningen and China Scholarship Council (CSC201708320268). The funders had no role in the study design, data collection and analysis, decision to publish or preparation of the manuscript.

## Author contributions

C.W., A.Z., M.G.N., R.K.W. and J.F. conceptualized and managed the study. L.C., V.C., M.J., I.C.L.v.d.M., A.V.V., A.K., R.G., T.S., M.O., L.A.B.J., J.H.W.R. and N.P.R. collected the samples and generated the data. L.C. analysed the data. L.C., V.C. and J.F. drafted the manuscript. All the authors reviewed and edited the manuscript.

## Competing interests

The authors declare no competing interests.
