## [Peer Review File · Nature Communications]

Reviewers' comments:

Reviewer #1 (Remarks to the Author):

Review of Chen et al., 'Gut microbial co-abundance networks identify functional hubs in inflammatory bowel disease and obesity'

This study by Chen et al. applies a correlation-based analysis to identify key organizational differences between the microbiome of four different cohorts: two population cohorts, an obese cohort, and an IBD cohort. A fundamental problem in defining microbiome 'state' is that most studies focus on a parts-list description of the microbiome: an enumeration of what is present and at what fraction the parts are present. This type of description misses critical information regarding microbiome structure, i.e. the interactions between members of the microbiota or functional repertoires of the microbiome. Chen et al. do a good job of identifying this problem and posing that considering interactions between microbial members or genetic elements of the microbiome is a worthwhile endeavor and one that should be considered.

As is said several times in the manuscript, the study conducted by Chen et al. is the 'largest metagenomics-based network analysis to date' and is therefore noble in cause. However, there are several issues that need to be addressed in order for the major conclusions presented (that there is a difference in network architecture between IBD, obesity, and 'normal') to be adopted by readers. This review will first address broad issues, then address specific ones that were evident while reading the manuscript.

General comments

The number of pathways (using the HUMAN2 pipeline) that satisfied the authors' threshold was on the order of 300. The number of possible pairwise interactions is therefore 3002 or $\sim 10^3$. Thus, to achieve adequate sampling for detecting statistically significant correlations, one needs at least 100 to 1000-fold the complexity of interactions, meaning that the number of people sampled would need to be on the order of 10⁷-10⁸, a far cry from the ~ 2500 people sampled in this study. Of paramount importance when assigning correlations to an under-sampled study is to make sure that the correlations identified are not spurious.

While sampling 107 people is not going to be a possibility in the near future, there are ways that others in the field have gotten around this problem. Leveraging longitudinal data has provided ways to study the stability of correlation networks to identify what features of organization are conserved and what are idiosyncratic or spurious to particular timepoints. Additionally, there are many ways to measure network organization including SparCC (the method the authors used), SPIEC-EASI, Singular Value Decomposition, t-SNE. Using any one of these on their own (as is the case in this manuscript) is placing too much emphasis on the fidelity of a particular approach, all of which have their own caveats, rules, and underlying mathematics. The authors make the point that they chose SparCC because of the suggestion of Weiss et al ('Correlation detection strategies in microbial data sets vary widely in sensitivity and precision', ISME Journal (2016)). In the two years since that paper has been written, newer more sophisticated methods have been employed to understand the organization of complex systems within and outside the field of microbiome science. It would be worth the authors' time to look into other methods of judging whether there are truly differences between the microbiomes of the cohorts using these other methods (SVD, t-SNE, SPIEC-EASI) and not solely trusting SparCC (which, in the hands of this reviewer, has produced mixed results at best).

Relatedly, the authors absolutely must define a null-model for correlation if assigning p-values to the results they observe, particularly in the limit that they are drastically under-sampled with respect to cohort size. Random matrix theory (RMT) approaches have demonstrated that non-random correlation structure can exist in finitely sampled datasets even when the matrix is comprised of shuffled data that maintains the underlying probability distributions. This is a substantial problem given the under-sampling evident in this study. A way to address this would be to answer the question, what would be a random model of correlation given ~ 2500 samples with ~ 300 pathways within this study? The lack of such a model creates unphysical (and

unbelievable) p-value results such as $P < 10^{-260}$ —a result that suggests either that a physical law (i.e. gravitation, laws of thermodynamics, Maxwell's equations, etc) has been identified from the data or, more likely, that the framework of the null hypothesis is invalid.

Given the limits in determining the validity of correlations in the paper, it is difficult to place faith in the interpretation of the results. It would be far more powerful to either (1) do an experiment to validate any of the findings, or (2) use other statistical methods that show a similar trend as those generated from the SparCC approach.

Specific comments

-Title: 'Functional Hubs' is an inaccurate wording. There is no evidence to suggest that the hubs themselves are 'functional'; merely that they differentiate between the statistically defined configurations of IBD and obesity

-Abstract: 'that might represent potential therapeutic targets for disease prevention and treatment'. This line is overused in microbiome science. In a paper where there are no experiments that reconfigure the microbiome or measure any effect on host physiology, it is a substantial stretch to say that any differentiating feature identified is not simply an epiphenomenon of a more fundamental underlying process underscoring important dynamical processes that have gone awry (i.e. host genetics and transcriptional patterns). Such statements need to be toned down across the field, and there is an opportunity to do this here.

-Introduction, line 65: Please avoid using words like 'strong'. This is a subjective criteria and, in the opinion of this reviewer, untrue. There is sparse evidence, at best, to suggest that the microbiome composition is related to development of diseases.

-Introduction, transition from Paragraph 1 to paragraph 2. The authors make a point of saying that interactions between ecological components are important to identify at the end of paragraph 1. Then in the beginning of paragraph 2 state that network inference tools have been developed. Why are statistical inferences valid substitutions for ecological interactions? There is a logical leap from needing to identify interactions to using statistics as a proxy for interactions. This needs to be explicated more.

-Results. SparCC is predicated on the log-transformation of variance. In this reviewer's experience, SparCC provides different results than SPIEC-EASI and Singular Value Decomposition. As stated above, if the authors performed other statistical techniques that are supposed to identify key 'features' in a complex system, how do the results compare to their current results?

-Results: Line 138. What are 'consistent' effects? A further description of what this entails would be helpful to understand what seems to be a powerful control in looking at a separate cohort of IBD

-Results: Lines 154-155. It would be worthwhile to perform PCA on the pathways outlined here to see if they separate cohorts. They should if the statistical significance holds true.

-Results: Lines 168-169. P values of $< 10^{-64}$ and 10^{-260} do not make sense. Please either reevaluate the null hypothesis or explain how these p-values are generated.

-Results: Line 170. There are 'xxx' and 'xx' words in the sentence. These need to be specified as these numbers are crucial to the results.

-Results: the use of HUMAnN2. What would happen if another pathway annotation scheme were used, i.e. mcSEED?

-Results: Lines 196-207. The functional link to physiology is specious. The co-abundant pathways are identified through statistical analysis of fecal samples; why should there be a correspondence

between what is observed in the feces with core metabolism in the organism?

-Methods: Line 571-572. It would be worth analyzing the longitudinal data of the iHMP to see what the stability of the co-abundant network is over time and through fluctuations in disease and recovery. IBD is a particularly salient use-case for looking at dynamics of the microbiome as patients go through phases of disease that vary in severity; thus each person can serve as their own 'control' so to speak.

Reviewer #2 (Remarks to the Author):

The manuscript by Chen et al., describes a large co-abundance network-based microbiota analysis in 4 different cohorts. Sample material was stool, which was handled identically between the cohorts and metagenomic sequences were obtained in a single centre. The co-abundance networks were reconstructed from species- and pathway-level information. The study claims that specific microbial co-abundance relationships are associated with the physiological (or pathological) state, however they also show a high degree of heterogeneity (64% at the pathway level). For the IBD cohort, effects were partially verified in an independent iHMP cohort. Cohort-specific edges were significantly enriched in the IBD and obesity cohorts and are described to be enriched in few hubs (obesity 1 pathway hub, IBD 5 species and 6 pathway hubs). The obesity hub is associated with allantoin degradation, the top pathway hubs for IBD was assigned to the reductive TCA cycle term.

The study is a large descriptive undertaking and makes use of existing metagenomic datasets from large cohorts. The employed algorithms and statistical approaches seem appropriate, however the manuscript lacks in my eyes the necessary clarity and scrutiny on physiological relevance of the findings.

The manuscript is written in a very technical style, rarely the approaches are bio-medically "translated". For a broader readership, I would strongly recommend re-writing the abstract, results (and discussion) section. The network lingo is not very instructive, I would also suggest to move the analytical scheme in abbreviated form into the main figures, so that one can follow the flow of analyses.

The entire study is based on features that are present in >20% in at least one of the cohorts. How did the authors define this number, what would happen if the cutoff is set to 5, 10 or 30 % ? Although the network analyses and figures are highly sophisticated, the clinical variables are only treated very superficially. There are networks specific to the "obesity" cohort, but clinically the BMI range is huge. Maybe I misunderstood, but have the authors tried to quantitatively model the co-abundance network with the BMI? If something is appearing in a cohort which samples high BMI individuals, shouldn't the same network properties also occur, if high BMI individuals are subsampled from the other cohorts ?

Also, the clinical attribute IBD is inappropriate if only used alone. The authors clearly must try to discriminate between CD/UC and to correlate their findings to clinical activity and co-medication. The stability assessment (p8, line 170ff.) refers to this to some degree, but is really unclear and vague.

Some strange technical typos: p8, line 170 "xxx species and xx pathway edges" ?, the references have strange page numbers (partially)

**Reviewer 1:**

This study by Chen et al. applies a correlation-based analysis to identify key
organizational differences between the microbiome of four different cohorts: two
population cohorts, an obese cohort, and an IBD cohort. A fundamental problem in
defining microbiome 'state' is that most studies focus on a parts-list description of the
microbiome: an enumeration of what is present and at what fraction the parts are
present. This type of description misses critical information regarding microbiome
structure, i.e. the interactions between members of the microbiota or functional
repertoires of the microbiome. Chen et al. do a good job of identifying this problem and
posing that considering interactions between microbial members or genetic elements of
the microbiome is a worthwhile endeavor and one that should be considered.

As is said several times in the manuscript, the study conducted by Chen et al. is the
'largest metagenomics-based network analysis to date' and is therefore noble in cause.
However, there are several issues that need to be addressed in order for the major
conclusions presented (that there is a difference in network architecture between IBD,
obesity, and 'normal') to be adopted by readers. This review will first address broad
issues, then address specific ones that were evident while reading the manuscript.

**Reply:** We thank the reviewer for their positive comments on the advances made in this
study. We have significantly revised the manuscript and added two separate result
sections "**Microbial co-abundance network in IBD**" (Line 151-232) and "**Microbial**
**co-abundance network in 3000B**" (Line 233-261). We believe that the results
regarding the difference in network architecture in IBD and obesity are now better
presented. We address the concerns raised by the reviewer in detail below.

**General comments**

The number of pathways (using the HUMAnN2 pipeline) that satisfied the authors'
threshold was on the order of 300. The number of possible pairwise interactions is
therefore 300^2 or $\sim 10^5$. Thus, to achieve adequate sampling for detecting statistically
significant correlations, one needs at least 100 to 1000-fold the complexity of
interactions, meaning that the number of people sampled would need to be on the order
of 10^7 - 10^8 , a far cry from the ~ 2500 people sampled in this study. Of paramount
importance when assigning correlations to an under-sampled study is to make sure that
the correlations identified are not spurious.

While sampling 10^7 people is not going to be a possibility in the near future, there are
ways that others in the field have gotten around this problem. Leveraging longitudinal

data has provided ways to study the stability of correlation networks to identify what
features of organization are conserved and what are idiosyncratic or spurious to
particular time points.

Additionally, there are many ways to measure network organization including SparCC
(the method the authors used), SPIEC-EASI, Singular Value Decomposition, t-SNE. Using
any one of these on their own (as is the case in this manuscript) is placing too much
emphasis on the fidelity of a particular approach, all of which have their own caveats,
rules, and underlying mathematics. The authors make the point that they chose SparCC
because of the suggestion of Weiss et al ('Correlation detection strategies in microbial
data sets vary widely in sensitivity and precision', ISME Journal (2016)). In the two
47 years since that paper has been written, newer more sophisticated methods have been
employed to understand the organization of complex systems within and outside the
field of microbiome science. It would be worth the authors' time to look into other
methods of judging whether there are truly differences between the microbiomes of the
cohorts using these other methods (SVD, t-SNE, SPIEC-EASI) and not solely trusting
SparCC (which, in the hands of this reviewer, has produced mixed results at best).

Relatedly, the authors absolutely must define a null-model for correlation if assigning p-
values to the results they observe, particularly in the limit that they are drastically
under-sampled with respect to cohort size. Random matrix theory (RMT) approaches
have demonstrated that non-random correlation structure can exist in finitely sampled
datasets even when the matrix is comprised of shuffled data that maintains the
underlying probability distributions. This is a substantial problem given the under-
sampling evident in this study. A way to address this would be to answer the question,
what would be a random model of correlation given ~2500 samples with ~300
pathways within this study? The lack of such a model creates unphysical (and
unbelievable) p-value results such as $P < 10^{-260}$ —a result that suggests either that a
physical law (i.e. gravitation, laws of thermodynamics, Maxwell's equations, etc) has
been identified from the data or, more likely, that the framework of the null hypothesis
is invalid.

Given the limits in determining the validity of correlations in the paper, it is difficult to
place faith in the interpretation of the results. It would be far more powerful to either
(1) do an experiment to validate any of the findings, or (2) use other statistical methods
that show a similar trend as those generated from the SparCC approach.

**Reply:** We thank the reviewer for pointing out several limitations of statistical inference
of microbiome networks in both ours and other studies, particularly three important
issues:

**1) Are the conclusions reproducible when applying another method?**

As suggested by the reviewer, we have now applied both SparCC and SPIEC-EASI for
network construction. SPIEC-EASI infers a network via an inverse covariance matrix
derived from compositional data after log-ratio transformation. SPIEC-EASI calculates
correlation coefficients based on partial correlation-based methods. In principle, SPIEC-
EASI can reduce indirect associations, but it can also make estimation of co-abundance
strength difficult to compare across different cohorts. Figure 1 below compares
correlation coefficients estimated by SparCC and SPIEC-EASI. Despite high correlation
between the two methods ($r > 0.81$, $P < 2.2 \times 10^{-16}$), the partial correlation correlations
estimated by SPIEC-EASI are indeed smaller than those estimated by SparCC. Of the
5,863 species and 56,519 pathway edges established by SparCC at $FDR < 0.05$ level, 3,454
(58.91%) and 43,355 (76.71%) were detected by SPIEC-EASI (Table 1).

Rebuttal Figure1. Correlation of species and pathway co-abundance strengths generated
by SparCC and SPIEC-EASI

We therefore consider these two methods to be complimentary and combined these two
methods in our revised study, i.e. we only consider microbial co-abundances that can be
detected by SparCC at $FDR < 0.05$ and by SPIEC-EASI (passed inverse covariance
selection model).

Rebuttal Table 1. Overlapped co-abundances between SparCC and SpiecEasi

		LLD	500FG	3000B	IBD
Species co-abundance	SparCC only	3931	2109	1368	3907
	SparCC + SpiecEasi	2604	1591	1107	2554
Pathway co-abundance	SparCC only	50121	44664	46744	47566
	SparCC + SpiecEasi	40699	37279	37886	37699

We have updated the method and manuscript accordingly. Notably, the general
 conclusion still holds. We found that 38.6% of species co-abundances and 64.3% of
 pathway co-abundances showed variable correlation strengths among our four cohorts,
 with 120 species and 1448 pathway edges showing cohort-specificity, mainly in IBD
 (113 IBD-specific species co-abundances and 1050 IBD-specific pathway co-
 abundances).

The Method section has been updated:

L445-448: *“To reduce indirect associations, we further applied SPIEC-EASI, which infers*
 *the microbial network underlying graphical model using the concept of conditional*
 *independence [38]. In this way, we obtained 3,454 species and 43,355 pathway co-*
 *abundances that were detectable by both methods (Fig 1).”*

**2) Addressing the power issue and leveraging longitudinal data to provide ways to**
 **study the stability of correlation networks and identify which features are**
 **conserved and which are idiosyncratic or spurious to particular time points.**

We fully agree with the reviewer that the current study is still under-sampled for
 comparing the number of interactions that we tested. We have discussed this limitation
 in the Discussion.

Line 324-327: *“However, we also acknowledge several limitations of our study. This is an*
 *in-silico network analysis based on correlation in bacterial abundance levels. Even with the*
 *largest sample size to date, our study is still undersized for making comparisons to the*
 *number of interactions assessed.”*

Following the reviewer’s valuable suggestion, we have now used longitudinal data of 77
 IBD patients from the integrative Human Microbiome Project (iHMP-IBD) to assess the
 stability of the correlation networks. Firstly, we replicated the IBD co-abundance
 networks using metagenomics data of the first sample collection from 77 iHMP-IBD
 participants. Out of the 2,090 and 37,106 IBD species and network co-abundances that
 can be assessed in the iHMP-IBD cohort, 1,705 (81.6%) species co-abundances and

27,886 (65.1%) of pathway co-abundances showed no difference in terms of their co-
abundance strength (Cochran-Q test $P > 0.05$). Then, we compared the IBD co-abundance
networks between the first and the last time points (~one year apart) in iHMP-IBD and
observed 90.6% and 99.6% replication for species and pathway co-abundances,
respectively (Cochran-Q test $P > 0.05$). These results are now discussed in the main text
and detailed results have been added to Tables S1 & S3.

L152-166: **“Replication of the IBD network in the iHMP-IBD cohort: Of the 2,554**
**species and 37,699 pathway co-abundances established in our IBD cohort, we were able to**
**assess 2,090 species co-abundances and 37,106 pathway co-abundances in 77 IBD**
**individuals from the integrative Human Microbiome Project (iHMP-IBD) [39]. In the**
**baseline samples of the iHMP-IBD cohort, 531 species co-abundances (25.4%) and 21,882**
**(59.0%) pathway co-abundance could be replicated at $P < 0.05$ (Tables S7-8) [39]. The**
**relatively low replication rate in species co-abundances is largely a power issue, as we also**
**observed that 1,705 (81.6%) species co-abundances and 24,165 (65.1%) pathway co-**
**abundances showed no significant difference in their co-abundance strengths between our**
**IBD cohort and the iHMP-IBD cohort (Cochran-Q test, $P > 0.05$, Fig S6, Tables S7-8). We then**
**compared the IBD networks between the first and last time points of the iHMP-IBD cohort**
**(~1 year apart) and replicated 90.6% of species co-abundances and 99.6% of pathway co-**
**abundances (Cochran-Q test, $P > 0.05$, Fig S6, Tables S7-8). This suggests that our**
**estimation of co-abundance strengths in IBD was largely replicable in a different cohort**
**and was stable across time.”**

The comparison is now shown in Supplementary figure 6.

**Figure S6.** Replication of the IBD
network using longitudinal data from
the iHMP-IBD cohort. We assessed the
replication rate of IBD co-abundances
in the iHMP-IBD cohort, as well as their
stability between the first and last time
points. Both the X- and Y-axis
represent the correlation coefficient of
co-abundances. Each dot represents
one co-abundance. Red dots represent
microbial co-abundances that show a
difference in their effect size between
the first and last time points at $P < 0.05$.

**3) P-value results such as $P < 10^{-260}$ are unbelievable. The authors absolutely must**
**define a null-model for correlation.**

We apologize for the confusion, the $P < 10^{-260}$ was not for co-abundance but for the
enrichment analysis of cohort-specific effects. We found a total of 1,448 cohort-specific
pathway co-abundances, with 1,050 of them related to IBD, 281 to the obesity cohort
and 117 to population-based cohort. Cohort enrichment was assessed using Fisher's
exact test, and the P value was estimated to be $P < 10^{-260}$. To make it clearer in the revised
manuscript, we have added Figure 3C & D (see below) to show the distribution of
cohort-specific co-abundances in different cohorts.

**Figure 3. C.** Pie chart of 120 cohort-specific species co-abundances showing the
proportion of specific co-abundances detected in each cohort. **D.** Pie chart of 1,448
cohort-specific pathway co-abundances showing the proportion of specific co-
abundances detected in each cohort.

For null-model of correlation, we applied SparCC default settings, i.e. we calculate a P-
value based on the distribution of correlation coefficients generated by using 100 times
permutation. The distribution of null-model correlation coefficients is shown in Figure
2, and the minimal P-value is close to 0.01 based on 100 times permutation. We further
calculated study-wise FDRs based on the permutation results (script available via:
[https://github.com/GRONINGEN-MICROBIOME-CENTRE/Groningen-](https://github.com/GRONINGEN-MICROBIOME-CENTRE/Groningen-Microbiome/tree/master/Projects/Microbial%20co-abundance%20network)
[Microbiome/tree/master/Projects/Microbial%20co-abundance%20network](https://github.com/GRONINGEN-MICROBIOME-CENTRE/Groningen-Microbiome/tree/master/Projects/Microbial%20co-abundance%20network)).

Rebuttal Figure 2. Distribution of null-model SparCC correlation coefficients generated
by 100 times permutation

Specific comments

-Title: 'Functional Hubs' is an inaccurate wording. There is no evidence to suggest that
the hubs themselves are 'functional'; merely that they differentiate between the
statistically defined configurations of IBD and obesity

**Reply:** We agree with this reviewer, now have changed the title to: "*Gut Microbial Co-*
*abundance Networks Show Specificity in Inflammatory Bowel Disease and Obesity*"

-Abstract: 'that might represent potential therapeutic targets for disease prevention and
treatment'. This line is overused in microbiome science. In a paper where there are no
experiments that reconfigure the microbiome or measure any effect on host physiology,
it is a substantial stretch to say that any differentiating feature identified is not simply
an epiphenomenon of a more fundamental underlying process underscoring important
dynamical processes that have gone awry (i.e. host genetics and transcriptional
patterns). Such statements need to be toned down across the field, and there is an
opportunity to do this here.

**Reply:** We thank this reviewer for pointing out the over-interpretation of results, we
have now changed the text to:

L49-52: "*Our study identifies several key species and pathways in IBD and obesity and*
*provides evidence that altered microbial abundances in disease can reflect their co-*
*abundance relationship, which expands our current knowledge regarding microbial*
*dysbiosis in disease.*"

-Introduction, line 65: Please avoid using words like 'strong'. This is a subjective criteria
and, in the opinion of this reviewer, untrue. There is sparse evidence, at best, to suggest
that the microbiome composition is related to development of diseases.

**Reply:** We thank the reviewer for pointing this out, we have now changed it to:

L59-62: "*In recent years, associations have been identified between gut microbiome*
*composition and the development of certain human diseases including diabetes ^{6,7},*
*cardiovascular disorders ^{8,9}, obesity ^{10,11} and chronic gastrointestinal disorders like*
*inflammatory bowel disease (IBD) ¹²⁻¹⁴*"

-Introduction, transition from Paragraph 1 to paragraph 2. The authors make a point of
saying that interactions between ecological components are important to identify at the

end of paragraph 1. Then in the beginning of paragraph 2 state that network inference
tools have been developed. Why are statistical inferences valid substitutions for
ecological interactions? There is a logical leap from needing to identify interactions to
using statistics as a proxy for interactions. This needs to be explicated more.

**Reply:** We thank the reviewer for pointing this out, we have now added:

L68-72: *“Enthusiasm has thus been rising to decipher these microbial interactions in order*
*to detect key microbes in health and disease* ^{23,24}. *One way of doing this is to create co-*
*abundance networks based on correlations, a method that has the potential to study*
*interactions between microbes and thereby generate hypotheses for experimental*
*validation at a later stage* ^{23,24}”

-Results. SparCC is predicated on the log-transformation of variance. In this reviewer’s
experience, SparCC provides different results than SPIEC-EASI and Singular Value
Decomposition. As stated above, if the authors performed other statistical techniques
that are supposed to identify key ‘features’ in a complex system, how do the results
compare to their current results?

**Reply:** We agree with the reviewer’s comment and have now applied both SparCC and
SPIEC-EASI. For details please see the answer above.

-Results: Line 138. What are ‘consistent’ effects? A further description of what this
entails would be helpful to understand what seems to be a powerful control in looking at
a separate cohort of IBD

**Reply:** We have changed “consistent” to “comparable”, i.e. they do not show
heterogeneity. Apart from the cross-sectional replication in the iHMP-IBD cohort, we
have now also added longitudinal replication by using the first and last time point
samples from 77 iHMP-IBD participants (~one year apart). Here we observed that,
indeed, microbial network in IBD were stable. This result has now been added to the
result section:

L162-166: *“We then compared the IBD networks between the first and last time points of*
*the iHMP-IBD cohort (~1 year apart) and replicated 90.6% of species co-abundances and*
*99.6% of pathway co-abundances (Cochran-Q test, $P>0.05$, Fig S6, Tables S7-8). This*
*suggests that our estimation of co-abundance strengths in IBD was largely replicable in a*
*different cohort and was stable across time.”*

-Results: Lines 154-155. It would be worthwhile to perform PCA on the pathways
outlined here to see if they separate cohorts. They should if the statistical significance
holds true.

**Reply:** We have now included the PCA plot of both microbial species and pathways. We
found that the four cohorts were largely overlapped, we also observe significant
differences in microbial species and pathway composition between cohorts (Wilcoxon
test $P < 0.05$).

This result has been added to the main text. Line 94-97: “Metagenomic data of the 2,379
participants from the four cohorts was processed using the same pipeline. Principle
coordinate analysis showed that microbial composition and functional profiles are largely
overlapped, although we observed a significant shift in species composition in the IBD
cohort (Fig S2).”

The PCoA plot has also been shown in the supplementary figure 2.

**Figure S2.** Principal component analysis of microbial species and pathways. A. PCoA
(Bray-Curtis distance matrix) of 134 species that are present in >20% of samples in at
least one cohort. B. PCA (Euclidean distance matrix) of 343 pathway that are present in
>20% of samples in at least one cohort. The Wilcoxon test was applied to assess
microbial compositional difference between cohorts.

-Results: Lines 168-169. P values of $<10^{-64}$ and 10^{-260} do not make sense. Please either
reevaluate the null hypothesis or explain how these p-values are generated.

**Reply:** We apologize for the confusion. These P-values are not for correlation. They are
P-values for cohort enrichment estimated by Fisher's exact test. We have now added the
pie charts in Fig. 3C&D to show the distribution of cohort-specific effects and have
further clarified this in the text.

L139-146: *"Interestingly, cohort-specific co-abundances were significantly enriched in the*
*disease cohorts compared to the population-based cohorts: 113 (94%) species co-*
*abundances and 1,050 (72%) pathway co-abundances were specifically related to the IBD*
*cohort (Fisher's test $P=1.2 \times 10^{-56}$ and $P < 10^{-260}$, respectively, Fig 3C-D) and 281 (19.4%)*
*pathway co-abundances were specifically related to the 3000B cohort (Fisher's test*
*$P=2.9 \times 10^{-29}$), as compared to only 3 species and 117 pathway co-abundance relationships*
*specific to the population-based cohorts LLD and 500FG (Fig 3C-D)."*

-Results: Line 170. There are 'xxx' and 'xx' words in the sentence. These need to be
specified as these numbers are crucial to the results.

**Reply:** We apologize for this inadvertent mistake. Now we have fixed it.

L144-146: *"as compared to only 3 species and 117 pathway co-abundance relationships*
*specific to the population-based cohorts LLD and 500FG (Fig 3C-D)."*

-Results: the use of HUMAnN2. What would happen if another pathway annotation
scheme were used, i.e. mcSEED?

**Reply:** We thank the reviewer for this suggestion. We acknowledge that knowledge of
microbial functionality is still limited. None of pathway annotation tools can give a
comprehensive picture of the microbial functional profile, and our analysis may be
biased due to annotation of HUMAnN2. Therefore, instead of re-doing all analysis using
mcSEED, we decide to discuss the limitation of our study. We sincerely hope that this
addresses the concerns of the reviewer.

L329: *"However, we also acknowledge several limitations of our study. This is an in-silico*
*network analysis based on correlation in bacterial abundance levels. Even with the largest*
*sample size to date, our study is still undersized for making comparisons to the number of*
*interactions assessed. In recent years, many different network tools have been developed to*
*tackle the statistical challenges in inferring networks for compositional data. In this study,*

*we applied two independent methods, SparCC and SpiecEasi, to establish microbial co-*
*abundance networks based on MetaPhlan and HUMAnN2 annotation. Our analysis can*
*thus be biased due to these annotation tools. Other annotation tools, e.g. mcSEED ⁶⁵, may*
*yield different pictures of microbial community and functional profile, thereby identifying*
*different co-abundance networks. Thus, such in-silico-based network inferences require*
*further functional validation. Although bacterial genes are believed to be expressed*
*uniformly ⁶⁶, previous studies have also shown that meta-transcription can exert dynamic*
*changes in response to environmental perturbations that cannot be detected at the*
*metagenome level ^{67,68}. Thus, in order to understand the microbial ecosystem in terms of*
*functional interaction in diseases, we need complementary approaches like meta-*
*proteomics and meta-metabolomics that provide a more direct readout of the functional*
*properties of the gut microbiome. Furthermore, the cross-sectional design of this study*
*makes it hard to assess the stability of our findings over time.”*

-Results: Lines 196-207. The functional link to physiology is specious. The co-abundant
pathways are identified through statistical analysis of fecal samples; why should there
be a correspondence between what is observed in the feces with core metabolism in the
organism?

**Reply:** We thank the reviewer for pointing this out and have removed that sentence.
Furthermore, we have revised the paragraph to avoid over-interpretation.

L249-261: *“When we compared microbial co-abundances in the 3000B to the other three*
*cohorts, we identified 281 pathway co-abundances that showed a significantly different*
*effect, i.e. obesity-specific co-abundances. One key pathway in obesity was degradation of*
*allantoin (PWY0-41, Fig 4B, Table S6), which showed obesity-specific co-abundance*
*relationships with 85 pathways. Allantoin is one of the active principles in various plants,*
*e.g. yams, and is found to enhance insulin secretion and lower plasma glucose ^{47,48}. Its*
*degradation product, oxamate, plays an inhibitory role in oxaloacetate/aspartate amino*
*acids ⁴⁹. In line with this, we found that the allantoin degradation pathway showed*
*stronger negative correlations with the biosynthesis pathways of oxaloacetate/aspartate*
*amino acids (including lysine, homoserine, methionine, threonine and isoleucine) and the*
*biosynthesis pathway of aspartate (PWY0-781, Fig 6), which were both positively*
*associated with fasting glucose level and negatively associated with fasting insulin level*
*($P < 0.05$, Table S15).”*

-Methods: Line 571-572. It would be worth analyzing the longitudinal data of the iHMP
to see what the stability of the co-abundant network is over time and through
fluctuations in disease and recovery. IBD is a particularly salient use-case for looking at
dynamics of the microbiome as patients go through phases of disease that vary in
severity; thus each person can serve as their own 'control' so to speak.

**Reply:** We thank the reviewer for this suggestion. We have now analyzed the
longitudinal data of the iHMP-IBD and compared the IBD co-abundance networks
between the first and the last sample collection from 77 iHMP-IBD participants (~one
365 year apart). Here we observed 90.6% and 99.6% replication rates for species and
366 pathway co-abundances, respectively. These results are now discussed in the main text,
and detailed results have been added into Tables S1 & S3. For more details please see
the answer above.

In addition, we have now performed systematic comparisons between IBD subtypes (UC
vs. CD), locations (colon vs. ileum) and disease activities (inflammation vs. no
inflammation) in our IBD cohort. Here we found that 16 species co-abundances were
related to disease subtype and 8 species co-abundances were related to disease location,
while 91, 24 and 3 pathway co-abundances were related to disease subtypes, location
and activity, respectively. The results have been added to the main text.

L167-183: *“Microbial networks of IBD in relation to disease characteristics. Previous
studies have shown that observed microbial abundance differences could be explained by
certain disease characteristics of IBD¹⁴. We therefore hypothesized that this could also be
the case for co-abundance relationships. We assessed whether IBD co-abundances
(including IBD co-abundances at FDR<0.05 and IBD-specific co-abundances) could be
related to the disease subtypes [ulcerative colitis (UC, n=189) vs. Crohn’s disease (CD,
n=276)], disease location [ileum (n=212) vs. colon (n=286)] and disease activity
[inflammation (n=121) vs. no inflammation (n=377)]. Most of the co-abundance
relationships were comparable between disease characteristics, and only a few showed
significant differences at FDR<0.05 (Fig S7, Tables S9-10), namely 16 species co-
abundances related to disease subtypes and 8 species co-abundances related to location.
For the pathway co-abundances, 91 were related to disease subtypes, 24 to location and 3
to activity (Cochran-Q test FDR<0.05, Fig S7). Out of these, five co-abundance relationships
were related to an important butyrate producer, *Faecalibacterium prausnitzii*, which
showed stronger co-abundance relationships in UC compared to CD. One example here was
the negative co-abundance relationship of *F. prausnitzii* with *Haemophilus parainfluenzae*,
a species known to have pathogenic properties⁴⁰.”*

Line 184- 192: “Microbial networks of IBD in relation to medication. We further tested
 whether drug usage can affect microbial co-abundance, as usage of antibiotics (20.0%)
 and proton pump inhibitors (PPIs, 26.5%) was higher in patients with IBD than in the
 general population cohorts (1.1% and 8.4%). Here we detected no significant difference in
 species co-abundances between antibiotic users and non-users (Cochran-Q test $FDR > 0.05$,
 Fig S7), while 1,049 out of 37,959 (3.7%) pathway co-abundance relationships showed
 statistically significant differences between PPI users and non-users, in particular related
 to the isoprene biosynthesis and methylerythritol phosphate pathways (Cochran-Q test
 $FDR < 0.05$, Fig S7, Table S10).”

The comparisons between disease sub-phenotypes and medication usages are also
 shown in the Figure S7 (see below).

**Figure S7.** IBD co-abundances in relation to sub-phenotypes. We assessed whether
 microbial co-abundances in IBD showed difference between IBD subtypes (UC vs. CD),
 disease activities (inflammation vs. no inflammation) and locations (ileum vs. colon) and
 with the usage of PPIs and antibiotics. Upper panel represents species co-abundances.
 Lower panel represents pathway co-abundances. Each dot represents one co-
 abundance. Red dots represent microbial co-abundances that show a difference in their
 effect size between sub-phenotypes at $FDR < 0.05$.

**Reviewer 2:**

The manuscript by Chen et al., describes a large co-abundance network-based
microbiota analysis in 4 different cohorts. Sample material was stool, which was
handled identically between the cohorts and metagenomic sequences were obtained in a
single centre. The co-abundance networks were reconstructed from species- and
pathway-level information. The study claims that specific microbial co-abundance
relationships are associated with the physiological (or pathological) state, however they
also show a high degree of heterogeneity (64% at the pathway level). For the IBD
cohort, effects were partially verified in an independent iHMP cohort. Cohort-specific
edges were significantly enriched in the IBD and obesity cohorts and are described to be
enriched in few hubs (obesity 1 pathway hub, IBD 5 species and 6 pathway hubs). The
obesity hub is associated with allantoin degradation, the top pathway hubs for IBD was
assigned to the reductive TCA cycle term.

The study is a large descriptive undertaking and makes use of existing metagenomic
datasets from large cohorts. The employed algorithms and statistical approaches seem
appropriate, however the manuscript lacks in my eyes the necessary clarity and scrutiny
on physiological relevance of the findings. The manuscript is written in a very technical
style, rarely the approaches are bio-medically “translated”. For a broader readership, I
would strongly recommend re-writing the abstract, results (and discussion) section.

**Reply:** We appreciate the reviewer’s suggestion. We have substantially revised the
manuscript, significantly reduced the technical description, and added more
interpretation regarding the biomedical relevance. In particular, we now include two
separate result sections on “**Microbial co-abundance network in IBD**” (Line 151-232)
and “**Microbial co-abundance network in 3000B**” (Line 233-261). The Discussion has
also been strengthened. We have also discussed on limitations of the current study. We
also believe that the readability has been improved to reach a wider audience.

The network lingo is not very instructive, I would also suggest to move the analytical
scheme in abbreviated form into the main figures, so that one can follow the flow of
analyses.

**Reply:** We thank the reviewer for this valuable suggestion, we have now switched the
analysis work flow (Figure S1) to main Figure 1.

The entire study is based on features that are present in >20% in at least one of the
 cohorts. How did the authors define this number, what would happen if the cutoff is set
 to 5, 10 or 30%?

**Reply:** The reviewer questioned the choice of 20% as a filter cutoff for species and
 pathways. Please note that there is no conventional threshold set in the field. Many
 microbial association studies, including many of our previous studies and the recent
 iHMP study (Lloyd-Price et al., Nature, 2019), chose to use 10% presence and/or at least

0.01% abundance level as their filter. However, these studies often link very sparse
 microbial data to rather complete metadata. In the current microbial network analysis,
 we have to link very sparse microbial data to itself. Moreover, the aim of our study was
 to not only construct microbial networks but also to compare networks between
 cohorts. The sample sizes of our four cohorts varied greatly, ranging from 1,135 in LLD
 to 298 in the obesity cohort. It is therefore important to ensure there are enough non-
 zero samples per cohort for reliable co-occurrence and co-abundance detection. We thus
 increased the cutoff to 20% to ensure sufficient data points for pair-wise correlation. At
 our cutoff, we identified 134 species and 343 pathways present in all the four cohorts
 with a minimal average abundance of 0.07%. Moreover, these species and pathways
 sufficiently captured the microbial composition, collectively accounting for, on average,
 86.9% of bacterial species and 99.9% of functional composition (please see also the
 compositionality analysis below). Furthermore, 91% of the common species (123 out of
 134) and 99% of the common pathways (340 out of 343) were also detected in the IBD
 cohort (n=77) from the iHMP-IBD project, which supports the robustness of the 20%
 cutoff.

Following the reviewer’s suggestion, we also checked microbial networks by applying a
 5%, 10% and 30% cutoff (see rebuttal Table 2 below). By applying different cutoffs, we
 observed that pathway co-abundance networks are comparable between different
 cutoffs, as they are less sparse than species data. We detected the most variable co-
 abundances for species at 20% cutoff. Thus, we have decided to continue using the 20%
 cutoff in our study.

Rebuttal Table 2: Number of co-abundances by different filtering cutoff.

	5%	10%	20%	30%
No. of species	226	174	134	101
Percentage of variable species co-abundances	16.0%	20.1%	38.6%	21.6%
No. of pathways	378	365	343	332
Percentage of variable pathway co-abundances	69.0%	65.1%	64.3%	70.2%

Although the network analyses and figures are highly sophisticated, the clinical
 variables are only treated very superficially. There are networks specific to the “obesity”
 cohort, but clinically the BMI range is huge. Maybe I misunderstood, but have the
 authors tried to quantitatively model the co-abundance network with the BMI? If
 something is appearing in a cohort which samples high BMI individuals, shouldn’t the

same network properties also occur, if high BMI individuals are subsampled from the
other cohorts?

**Reply:** We thank the reviewer for this valuable suggestion. To replicate microbial
networks in 3000B, we selected 134 obese individuals from the LLD cohort with
matched age and BMI. For the replication rate, we considered a co-abundance to be
replicable if the estimated correlation coefficient was comparable between 3000B and
the replication cohort (Cochran-Q test heterogeneity test $P > 0.05$). For 1,107 species and
37,886 pathway co-abundances detected in the 3000B cohort, 991 (89.5%) species co-
abundance and 32,963 (87.0%) pathway co-abundance show no difference in the
replication cohort, suggesting our findings are largely replicable. We have now added
this to the result section.

Line238: *“Replication of 3000B network in LLD obese individuals. 1,107 species and*
*37,886 pathway co-abundances were detected in the 3000B cohort (Fig 2). These*
*estimated co-abundance strengths were largely replicable in 134 obese individuals with*
*matched age and BMI from the LLD cohort, with 991 (89.5%) species co-abundances and*
*32,963 (87.0%) pathway co-abundances showing no difference (Cochran-Q test $P > 0.05$, Fig*
*S8, Tables S13-14).”*

Moreover, the comparison has also been shown in the supplementary figure 8

**Figure S8.** Replication of obesity network in 134
obesity individuals from the LLD cohort. The
comparisons of co-abundance strengths in terms of
correlation coefficients in the 3000B cohort and in 134
obesity individuals from the LLD cohort with similar
ages and BMIs. X-axis represents the estimated
correlation coefficients in the 3000B cohort. Y-axis
represents the estimated correlation coefficients in
obese individuals from the LLD cohort. Upper panel
represents species co-abundances. Lower panel
represents pathway co-abundances. Each dot
represents one co-abundance. Red dots represent
microbial co-abundances that show a difference in
their effect size between first and last time points at
$P < 0.05$.

In addition, we further assessed the relevance of microbial networks in the obesity
cohort to obesity-related diseases, namely atherosclerosis and type-2-diabetes.

Line 240-248: "Microbial networks in relation to obesity-related diseases. The 3000B
cohort was set up to study cardiovascular disease in obese individuals, including 139
patients with atherosclerotic plaque and 159 obese controls. In addition, 35 3000B
participants had diabetes. Here we observed only three species co-abundances related to
cardiovascular disease, with all three showing stronger co-abundances in patients with
plaque than in patients without (Cochran-Q test $FDR < 0.05$, Fig S9, Tables S13-14). These
were positive co-abundances between *Dorea longicatena* and *Dorea formicigenerans* and
negative co-abundances of *Lachnospiraceae* bacterium 9.1.43BFAA with *Coprococcus*
*comes* and *Dorea longicatena*."

These comparisons are also presented in the Figure S9 (see below).

**Figure S9.** Obesity co-abundances in relation to phenotypes. We further assessed
whether microbial co-abundances in 3000B showed difference between patients with
and without diabetes and atherosclerotic plaque. Upper panel represents species co-
abundances. Lower panel represents pathway co-abundances. Each dot represents one
co-abundance. Both the X- and Y-axes represent correlation coefficient of co-
abundances. Red dots represent microbial co-abundances that show a difference in their
effect size between subtypes at $FDR < 0.05$.

Also, the clinical attribute IBD is inappropriate if only used alone. The authors clearly
must try to discriminate between CD/UC and to correlate their findings to clinical

activity and co-medication. The stability assessment (p8, line 170ff.) refers to this to
some degree, but is really unclear and vague.

**Reply:** We apologize for the unclear description. We have now systematically assessed
the microbial networks of IBD in relation to disease subtypes (CD vs UC), location (colon
vs ileum) and activities (inflammation vs no inflammation). We also assessed their
relevance to medication use, especially of antibiotics and proton pump inhibitors. We
have now added a more detailed description into the two separate result sections.

L167-183: *“Microbial networks of IBD in relation to disease characteristics. Previous*
*studies have shown that observed microbial abundance differences could be explained by*
*certain disease characteristics of IBD*¹⁴. *We therefore hypothesized that this could also be*
*the case for co-abundance relationships. We assessed whether IBD co-abundances*
*(including IBD co-abundances at FDR<0.05 and IBD-specific co-abundances) could be*
*related to the disease subtypes [ulcerative colitis (UC, n=189) vs. Crohn’s disease (CD,*
*n=276)], disease location [ileum (n=212) vs. colon (n=286)] and disease activity*
*[inflammation (n=121) vs. no inflammation (n=377)]. Most of the co-abundance*
*relationships were comparable between disease characteristics, and only a few showed*
*significant differences at FDR<0.05 (Fig S7, Tables S9-10), namely 16 species co-*
*abundances related to disease subtypes and 8 species co-abundances related to location.*
*For the pathway co-abundances, 91 were related to disease subtypes, 24 to location and 3*
*to activity (Cochran-Q test FDR<0.05, Fig S7). Out of these, five co-abundance relationships*
*were related to an important butyrate producer, Faecalibacterium prausnitzii, which*
*showed stronger co-abundance relationships in UC compared to CD. One example here was*
*the negative co-abundance relationship of F. prausnitzii with Haemophilus parainfluenzae,*
*a species known to have pathogenic properties*⁴⁰.”

Line 184- 192: *“Microbial networks of IBD in relation to medication. We further tested*
*whether drug usage can affect microbial co-abundance, as usage of antibiotics (20.0%)*
*and proton pump inhibitors (PPIs, 26.5%) was higher in patients with IBD than in the*
*general population cohorts (1.1% and 8.4%). Here we detected no significant difference in*
*species co-abundances between antibiotic users and non-users (Cochran-Q test FDR>0.05,*
*Fig S7), while 1,049 out of 37,959 (3.7%) pathway co-abundance relationships showed*
*statistically significant differences between PPI users and non-users, in particular related*
*to the isoprene biosynthesis and methylerythritol phosphate pathways (Cochran-Q test*
*FDR<0.05, Fig S7, Table S10).”*

These comparisons are also presented in the supplementary figure 7 (see below).

**Figure S7.** IBD co-abundances in relation to sub-phenotypes. We assessed whether
microbial co-abundances in IBD showed differences between IBD subtypes (UC vs. CD),
disease activities (inflammation vs. no inflammation) and locations (ileum vs. not-ileum
(colon)) and with the usage of PPI and antibiotics. Upper panel represents species co-
abundances. Lower panel represents pathway co-abundances. Each dot represents one
co-abundance. Red dots represent microbial co-abundances that show a difference in
their effect size between sub-phenotypes at $FDR < 0.05$.

Some strange technical typos: p8, line 170 “xxx species and xx pathway edges” ?

**Reply:** We apologize for this inadvertent mistake. Now we have fixed it.

L144-146: “as compared to only 3 species and 117 pathway co-abundance relationships
specific to the population-based cohorts LLD and 500FG (Fig 3C-D).”

the references have strange page numbers (partially)

**Reply:** The references have been thoroughly checked and we have now fixed the page
numbers.

REVIEWERS' COMMENTS:

Reviewer #1 (Remarks to the Author):

All of my comments have been satisfactorily answered.

Reviewer #2 (Remarks to the Author):

The authors have responded to most of my points , I find the paper much improved.

Minor:

1. In the IBD cohort , I would demand a formal cohort description in form of a table (main stratum: UC /CD , substrata: inflammatory activity, medication including IBD-specific medication, age distribution, disease location) Currently, the way it is described it its confusing for a clinical reader, as all categories are independent, i.e. subtype or inflammatory activity or medication.
2. Antibiotics / PPI are important , but I would request a formal correlation analysis of networks to IBD specific medication (i.e. naive vs cortisone usage , naive vs. immunosuppressants and naive vs. biologicals) as several papers have pinpointed microbiome states and response to therapies (e.g. vedo or IFX).

Reviewer 2:

Minor:

1. In the IBD cohort, I would demand a formal cohort description in form of a table (main stratum: UC /CD , substrata: inflammatory activity, medication including IBD-specific medication, age distribution, disease location) Currently, the way it is described it its confusing for a clinical reader, as all categories are independent, i.e. subtype or inflammatory activity or medication.

Reply: We thank the reviewer for this suggestion. We have now added a supplementary table to summarize the clinical characterization of the IBD and the 300OB cohorts.

Supplementary Table 1: Summary of sub-phenotypes in the IBD and obesity.

Phenotypes	IBD (n = 496)		
	CD (n = 276)	UC (n = 189)	IBDU (n = 31)
Age mean (range)	41.2 (18- 81)	46.6 (19- 82)	44.2 (19- 76)
Disease location			
Colon n (%)	59 (22)	189 (100)	31 (100)
Ileum n (%)	92 (35)	0 (0)	0 (0)
Both n (%)	112 (43)	0 (0)	0 (0)
Active disease n (%)	69 (25)	46 (25)	6 (24)
Antibiotics yes (%)	58 (21)	32 (17)	5 (16)
IBD-medication			
Mesalazines yes(%)	25 (9)	123 (65)	23 (74)
Steroids yes (%)	46 (17)	31 (16)	4 (13)
Immunosuppressants yes (%)	129 (47)	65 (34)	7 (23)
Anti-TNFalpha yes (%)	101 (37)	19 (10)	3 (10)
Thiopurines yes (%)	97 (35)	52 (28)	4 (13)
Other biologicals yes (%)	3 (1)	0 (0)	0 (0)
Other medications			
ACE-inhibitor yes (%)	10 (4)	10 (5)	4 (13)
angII-receptor antagonist yes (%)	4 (1)	5 (3)	1 (3)
Beta-blockers yes (%)	15 (5)	10 (5)	6 (19)
Bisphosphonates yes (%)	6 (2)	5 (3)	0 (0)
Iron supplementation yes (%)	7 (3)	6 (3)	0 (0)
Folic acid yes (%)	26 (9)	1 (1)	2 (6)
Laxatives yes (%)	20 (7)	6 (3)	3 (10)
Metformin yes (%)	2 (1)	4 (2)	1 (3)
NSAID yes (%)	13 (5)	4 (2)	4 (13)
Opiat yes (%)	19 (7)	1 (1)	1 (3)
Platelet aggregation inhibitor yes (%)	12 (4)	11 (6)	3 (10)
PPI yes (%)	66 (24)	28 (15)	7 (23)
SSRI-antidepressant yes (%)	5 (2)	2 (1)	2 (6)
Statin yes (%)	9 (3)	14 (7)	3 (10)
Thiazide diuretic yes (%)	6 (2)	9 (5)	1 (3)
300OB (n = 298)			
Age mean (range)	67.1 (54- 80)		
Diabetes yes (%)	35 (12)		
Atherosclerotic plaque yes (%)	139 (47)		

2. Antibiotics / PPI are important, but I would request a formal correlation analysis of networks to IBD specific medication (i.e. naive vs cortisone usage, naive vs. immunosuppressants and naive vs. biologicals) as several papers have pinpointed microbiome states and response to therapies (e.g. vedo or IFX).

Reply: We thank the reviewer for the suggestion. Unfortunately, we are not able to perform the requested analyses for three reasons. Firstly, we lack treatment-naïve patients as controls because the IBD cohort in the present study does not contain any treatment-naïve patients. All IBD patients are enrolled at the University Medical Center Groningen, which is a tertiary hospital. Patients entering this tertiary hospital are already under IBD treatment at the first line or in a secondary hospital. Secondly, we cannot disentangle whether the observed effects are specific to the drug or the IBD subtype. Some commonly used IBD medications are subtype-specific, mesalazines, for example, are mostly used for the treatment of ulcerative colitis (see also Supplementary Table 1). Thirdly, we do not have enough power to do these analyses in some drugs, such as biologicals. The number of drug users is also very small (see Supplementary Table 1).